# TRUST THE UNCERTAIN TEACHER: DISTILLING DARK KNOWLEDGE VIA CALIBRATED UNCERTAINTY

## ABSTRACT

The core of knowledge distillation lies in transferring the teacher's rich '*dark knowledge*'—subtle probabilistic patterns that reveal how classes are related and the distribution of uncertainties. While this idea is well established, teachers trained with conventional cross-entropy often fail to preserve such signals. Their distributions collapse into sharp, overconfident peaks that appear decisive but are in fact brittle, offering little beyond the hard label or subtly hindering representation-level transfer. This overconfidence is especially problematic in high-cardinality tasks, where the nuances among many plausible classes matter most for guiding a compact student. Moreover, such brittle targets reduce robustness under distribution shift, leaving students vulnerable to miscalibration in real-world conditions. To address this limitation, we revisit distillation from a distributional perspective and propose Calibrated Uncertainty Distillation (CUD), a framework designed to make dark knowledge more faithfully accessible. Instead of uncritically adopting the teacher's overconfidence, CUD encourages teachers to reveal uncertainty where it is informative and guides students to learn from targets that are calibrated rather than sharpened certainty. By directly shaping the teacher's predictive distribution before transfer, our approach balances accuracy and calibration, allowing students to benefit from both confident signals on easy cases and structured uncertainty on hard ones. Across diverse benchmarks, CUD yields students that are not only more accurate, but also more calibrated under shift and more reliable on ambiguous, long-tail inputs.

## 1 INTRODUCTION

LLMs such as GPT-4 (OpenAI, 2023), PaLM (Chowdhery et al., 2022), LLaMA (Touvron et al., 2023), and Gemini (Gemini Team (Google), 2023) deliver state-of-the-art results across reasoning, dialogue, and code generation, yet they impose heavy latency, memory, and energy costs at inference time. Among compression techniques including pruning and quantization, knowledge distillation (KD) (Hinton et al., 2015) remains uniquely attractive because it transfers a teacher's *predictive distribution*, not just argmax labels, thereby capturing inter-class structure, inductive biases, and task-dependent uncertainty that smaller models would not infer from hard labels alone(Huang et al., 2022; Moslemi et al., 2024). The central hypothesis of this work is that, when appropriately preserved and transferred, the teacher's distribution better approximates the latent true label distribution over plausible classes (Dong et al., 2024), producing students that generalize more reliably.

However, conventional supervised training often yields over-confident teacher(Wei et al., 2022; El Baida et al., 2025) whose probabilities collapse onto a single class, suppressing precisely the soft structure that makes KD effective. Post-hoc temperature scaling, which adjusts prediction sharpness with a global scalar after training, only rescales logits and cannot recover information that was never represented or was erased by overfitting (Guo et al., 2017). Regularizers such as label smoothing and confidence penalties may reduce calibration error, yet they can also blur informative classwise geometry that students should imitate (Müller et al., 2019; Pereyra et al., 2017). Much of the KD literature compensates on the student side while keeping the teacher fixed, for example via KL matching with a fixed temperature, intermediate feature hints, or self-distillation (Zhang et al., 2019; Mobahi et al., 2020). These strategies are valuable, but they neither prevent distribution collapse in the teacher nor selectively filter teacher errors during transfer, leaving performance on borderline and long-tail examples especially fragile.

We take a distributional perspective on KD and argue that two properties are jointly required for strong downstream students. First, the *teacher* should retain informative uncertainty, allocating non-trivial probability mass to semantically related alternatives where the task is intrinsically ambiguous. Second, the *transfer* should be calibrated so that the student imitates informative relations while attenuating misleading teacher peaks that arise from over-confidence or spurious correlations. Intuitively, KD works best when the teacher behaves like a well-calibrated Bayesian posterior on easy and hard inputs alike, and when the student is encouraged to inherit that calibrated structure rather than the teacher's certainty alone. We define these characteristics in problem formulation section.

This view aligns with the empirical regularities observed across the modalities. In fine-grained classification and language understanding, soft targets consistently improve sample efficiency by encoding *relative* similarities between classes that hard labels cannot convey. When the data distribution shift, students distilled from calibrated teachers achieve lower expected calibration error and maintain more stable confidence on near-OOD inputs. This stability leads to improved selective prediction and more favorable accuracy–rejection trade-offs(Guo et al., 2017; Mishra et al., 2023). In resource-constrained deployments, such as on-device assistants (Chi et al., 2025) and retrieval-augmented pipelines where latency budgets are tight (Jiang et al., 2025), students that inherit calibrated uncertainty achieve competitive accuracy while maintaining predictable confidence estimates, a property that downstream systems can exploit for routing, abstention, and cost-aware ensembling.

Our study operationalizes this distributional stance without relying on architectural tricks or task-specific heuristics. We focus on *why* preserving and selectively transferring dark knowledge is necessary and *when* it pays off, rather than on implementation details. Empirically, we show that students trained under this lens achieve higher accuracy at fixed compute, improved robustness on ambiguous and long-tail slices, and better probabilistic calibration. The results support the thesis that effective KD should prioritize calibrated structure over sharpened certainty, bringing the student closer to the true label distribution while avoiding the teacher's pitfalls.

## 2 Problem Formulation

### 2.1 Preliminaries and Limitations of Conventional KD

Let $\mathcal{X}$ be the input space and $\mathcal{Y} = \{1, \ldots, C\}$ the label set. A teacher model induces a predictive distribution $p_T(\cdot \mid x) \in \Delta^{C-1}$ for $x \in \mathcal{X}$, and a student model with parameters $\theta_S$ induces $p_S(\cdot \mid x; \theta_S) \in \Delta^{C-1}$. Conventional knowledge distillation (KD) trains the student to match the teacher (optionally combined with ground-truth supervision):

$$\min_{\theta_S} \ \mathbb{E}_{x \sim (\mathcal{D})}\Big[\mathcal{L}_{\mathrm{KD}}\big(p_S(\cdot \mid x; \theta_S), \tilde{p}_T(\cdot \mid x)\big)\Big] \ + \ \lambda \, \mathbb{E}_{(x,y) \sim \mathcal{D}}\big[\mathcal{L}_{\mathrm{CE}}\big(p_S(\cdot \mid x), y\big)\big], \qquad (1)$$

where $\tilde{p}_T(\cdot \mid x)$ denotes the *calibrated* teacher target (usually just a temperature scaled version of $p_T$), and $\mathcal{D}$ is the labeled set. In practice, however, maximum likelihood training often produces an over confident $p_T$, so that even such temperature scaled targets can remain over sharpened, erasing relational "dark knowledge" and amplifying erroneous peaks that are then propagated to $p_S$.

### 2.2 Guiding Principles for Calibrated Distillation

We posit that KD can be simultaneously *robust* and *informative* when the following conditions hold.

**C1: Difficulty-aware uncertainty.** For ambiguous or challenging inputs, the teacher should preserve uncertainty correspond to task difficulty rather than collapsing probability mass onto a single class. Concretely, the teacher should retain non-trivial probability on semantically related alternatives, thereby encoding relational cues that are essential for generalization.

**C2: Calibrated, selective imitation.** The student should not blindly replicate the teacher's distribution. Instead, it should selectively imitate the teacher's informative relational structure while suppressing spurious over-confident peaks arising from teacher errors. This preserves the benefits of the teacher's nuanced knowledge without propagating its mistakes.

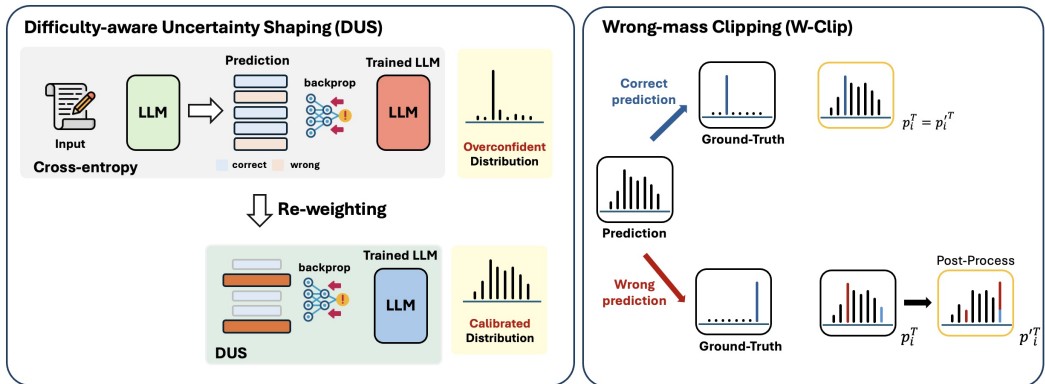

Figure 1: Illustration of the main framework. Left: Difficulty-aware Uncertainty Shaping (DUS) adjusts weights based on the correctness of the model's predictions, reducing overconfidence and encouraging a more calibrated probability distribution. Right: Wrong-mass Clipping (W-Clip) modifies the probability assigned to incorrect classes when the model makes a wrong prediction, providing a more stable and reliable distribution for the student.

### 2.3 Calibration as a Constraint-based Reformulation

Motivated by the above requirements, we recast knowledge distillation problem as a teacher calibration problem: we seek a calibration operator $\mathcal{C}$ such that, for each input $x$, the calibrated target $\tilde{p}_T(\cdot \mid x) = \mathcal{C}(p_T(\cdot \mid x), x)$ is obtained by projecting the raw teacher distribution onto the set of distributions that satisfy the following constraints R1 and R2:

**R1: Uncertainty with semantic support (for difficulty-aware uncertainty).** Maintain difficulty-appropriate concentration and guarantee mass over a stable neighborhood of semantically related classes:

$$h_{\min}(x) \ \leq \ H\big(\tilde{p}_T(\cdot \mid x)\big) \ \leq \ h_{\max}(x), \qquad \sum_{k \in \mathcal{N}(y;x)} \tilde{p}_T(k \mid x) \ \geq \ \alpha(x).$$

Here $H(\cdot)$ is an uncertainty measure (e.g., entropy), and $\mathcal{N}(y;x)$ is a semantics-preserving neighborhood of the true label $y$ (e.g., defined via taxonomies, embeddings, or empirical confusion).

**R2: Wrong-mass budget (for selective imitation).** Bound probability assigned to evidently incompatible classes:

$$\sum_{k \in \mathcal{W}(x,y)} \tilde{p}_T(k \mid x) \ \leq \ \varepsilon(x),$$

where $\mathcal{W}(x,y)$ denotes classes known to be inconsistent with $(x,y)$ (e.g., via rules, weak labels, or high-confidence constraints).

### 2.4 Projection to Calibrated Targets

Among all distributions satisfying these constraints, we select the one closest to the original teacher in a structure-preserving sense:

$$\tilde{p}_T \ \in \ \arg\min_{q \in \Delta^{C-1}} \ \mathrm{Dist}\big(q, \, p_T(\cdot \mid x)\big) \quad \text{s.t.} \quad \text{R1, R2,} \tag{2}$$

where $\mathrm{Dist}$ is a structure-sensitive divergence (e.g., symmetrized KL or Wasserstein). Coupled with the training objective in (1), this projection emphasizes calibrated pairwise ratios and ordinal relations rather than brittle peak sharpening. We prove in Appendix A that the optimization problem in (2) admits a unique global minimizer without spurious local minima.

# 3 CALIBRATED UNCERTAINTY DISTILLATION

While Eq. (2) establishes the theoretical ideal for target calibration, solving this constrained optimization problem directly for every training sample is computationally intractable. To bridge this gap, we propose Calibrated Uncertainty Distillation (CUD) designed to *relax* the reformulated constraints (R1, R2) in a practical and modular way. Specifically, our practical modules, DUS and W-Clip efficiently approximates the optimal solution: (i) **DUS**, which reshapes the teacher's confidence to counter overconfidence while retaining sharp predictions on easy cases, and (ii) **W-Clip**, which adjusts target distributions to suppress spurious peaks without disturbing relational structure. Together, these modules make the constraints operational and yield calibrated soft targets that improve knowledge transfer.

## 3.1 DIFFICULT-AWARE UNCERTAINTY SHAPING

DUS modifies the teacher distribution to provide uncertainty where it is most useful. The key idea is to *increase entropy selectively*: ambiguous or misclassified inputs are encouraged to carry higher uncertainty, while confident and correct predictions remain sharp. Concretely, we aim to raise entropy only when the teacher is uncertain or wrong—satisfying the lower bound $h_{\min}(x)$—and avoid unnecessary blurring on confident, correct cases to respect the upper bound $h_{\max}(x)$ and maintain class geometry.

Let $p_T(\cdot \mid x)$ be the teacher distribution and $p_T^y \triangleq p_T(y \mid x)$ for ground-truth $y$. We fine-tune the teacher with a **focal-entropy** objective that down-weights trivial cases and rewards non-degenerate uncertainty on hard or mispredicted inputs:

$$\mathcal{L}_{\text{teacher}} = \lambda_{\text{CE}}\big(-\log p_T^y\big) + \lambda_{\text{F}}\big(-\alpha_y(1-p_T^y)^\gamma \log p_T^y\big) - \lambda_{\text{H}}\, w(x)\, H\big(p_T(\cdot \mid x)\big), \qquad (3)$$

where $\alpha_y \in (0, 1]$, $\gamma \geq 0$, and $H(\cdot)$ denotes entropy. The difficulty gate

$$w(x) = \mathbb{1}\{p_T^y < \tau\} + \rho\,(1 - p_T^y)^\beta$$

activates entropy shaping only when the teacher is wrong or low-confidence. The focal factor $(1 - p_T^y)^\gamma$ reduces gradients on trivially correct examples ($p_T^y \approx 1$) and emphasizes hard/mispredicted ones ($p_T^y$ small); combined with the gated entropy reward $w(x)\, H(\cdot)$, this selectively increases entropy where it improves distillation (meeting $H(\tilde{p}_T) \geq h_{\min}(x)$) while leaving easy cases sharp (implicitly respecting $h_{\max}(x)$) without resorting to global label smoothing.

## 3.2 W-CLIP: WRONG-MASS CLIPPING

W-Clip calibrates the transfer by reallocating probability mass from clearly wrong classes. Instead of reshaping the full distribution, W-Clip applies a budgeted correction that suppresses an overconfident top-1 error when it conflicts with the ground truth. Formally, we approximate the wrong-mass constraint $\sum_{k \in \mathcal{W}(x,y)} \tilde{p}_T(k \mid x) \leq \varepsilon(x)$ by suppressing only the wrong top-1 class under a small, controllable budget.

Let $k^\star = \arg\max_k p_T(k \mid x)$ and suppose $k^\star \neq y$. We define two gated amounts

$$\delta_{\text{budget}}(x) \triangleq \eta\, p_T(k^\star \mid x), \qquad (4)$$

$$\delta_{\text{margin}}(x) \triangleq m\,\big(p_T(k^\star \mid x) - p_T(y \mid x)\big), \qquad (5)$$

where $\eta \in (0, 1)$ caps the total removable mass (the *wrong-mass budget*) and $m \in (0, 1]$ scales suppression with the teacher's over-confidence margin. We then set

$$\delta(x) = \min\{\delta_{\text{budget}}(x),\ \delta_{\text{margin}}(x)\}, \qquad (6)$$

$$\tilde{p}_T(y \mid x) = p_T(y \mid x) + \delta(x), \qquad \tilde{p}_T(k^\star \mid x) = p_T(k^\star \mid x) - \delta(x), \qquad (7)$$

$$\tilde{p}_T(k \mid x) = p_T(k \mid x) \quad \text{for } k \notin \{y, k^\star\}. \qquad (8)$$

If $k^\star = y$, we simply take $\tilde{p}_T = p_T$.

The budget term (4) directly implements the constraint that only a limited fraction of wrong-class mass can remain, while the margin term (5) focuses corrections on strongly over-confident mistakes. Keeping all non-involved classes unchanged preserves pairwise odds (dark knowledge), so the student inherits informative relations without propagating the teacher's spurious peak.

## 3.3 STUDENT DISTILLATION WITH CALIBRATED TARGETS

Given the calibrated targets $\tilde{p}_T(\cdot \mid x)$, the student minimizes a temperatured KL (plus optional CE on labeled data):

$$
\min_{\theta_S} \; \mathbb{E}_x \Big[ \lambda_{\mathrm{KD}} \, \tau^2 \, \mathrm{KL}\big(\tilde{p}_T^{(\tau)}(\cdot \mid x) \, \| \, p_S^{(\tau)}(\cdot \mid x; \theta_S)\big) \Big]
$$
$$
+ \; \lambda_{\mathrm{CE}} \, \mathbb{E}_{(x,y)} \big[ -\log p_S(y \mid x) \big], \tag{9}
$$

where $q^{(\tau)} = \mathrm{softmax}(\log q/\tau)$ and $\tau > 0$. This transfers the calibrated structure rather than brittle peaks; unlabeled examples contribute only the KD term, supporting semi-supervised settings.

## 3.4 OPERATIONAL REALIZATION OF CONSTRAINTS

In Section 2, we formulated target calibration as a constrained projection problem. Here, we connect our practical modules to this framework by operationalizing the abstract sets $N(y; x)$ and $W(x, y)$ for the constraints R1 and R2.

**Operationalizing Constraint Sets.** To render the projection tractable, we instantiate these sets based on the principal modes of teacher failure:

**1. Operational Wrong-Mass Set ($W_{op}$):** We define $W_{op}(x, y) = \{k^*\}$ strictly when the top-1 prediction $k^*$ contradicts the ground truth ($k^* \neq y$), and $\emptyset$ otherwise. This definition targets the dominant source of semantic misalignment identified in R2.

**2. Operational Neighborhood Set ($N_{op}$):** Rather than explicitly defining a semantic neighborhood, we operationalize R1 implicitly. Maximizing entropy via DUS widens the distribution, satisfying the neighborhood mass requirement without pre-defined similarity graph.

**Approximating the Optimal Projection.** As proven in Appendix A, the optimal solution $q^*$ to the constrained projection is an *exponential tilting* of the teacher distribution. However, computing the partition function for this tilt is computationally expensive. We demonstrate in Appendix A.3 that **W-Clip** constitutes a *first-order Taylor approximation* of this exponential tilt around $\nu = 0$. Specifically, the linear subtraction in W-Clip recovers the analytical solution ($e^{-\nu} \approx 1 - \nu$) for conservative corrections. Similarly, **DUS** acts as a regularizer that pre-conditions the teacher to satisfy the entropy bounds of R1. Thus, CUD replaces the intractable iterative projection with efficient $O(1)$ arithmetic operations grounded in information geometry.

# 4 EXPERIMENTAL RESULTS

## 4.1 TRAINING SETUP

**Models & Protocol.** We structure evaluation in two parts. **(Teacher-side)** We first train a standard *teacher*—`bert-base-uncased` (12 layers, 768 hidden)—and compare its performance against our distillation-friendly teacher (DUS) under the same data splits. **(Student-side)** We then distill from the trained teacher and compare students across KD methods, using two capacity variants to probe different architectural gaps: **Mini** (4 layers, 256 hidden), which changes both depth and width so feature-based distillation is inapplicable, and **Small** (6 layers, 768 hidden), which preserves hidden size and thus admits feature-KD baselines; our method uses calibrated response targets in both cases. All results are reported as the mean and variance computed over three independent runs.

**Hyperparameters & Rationale.** While our method introduces several hyperparameters, most show low sensitivity; we tuned them once on the Banking dataset and used the same global values for all tasks, avoiding per-dataset hyperparameter engineering (see Appendix B). The key distillation parameters include the entropy reward $\lambda_{\mathrm{H}} = 0.1$, the focal parameters $\gamma = 10$ and $\alpha_y = 1$, and the entropy based gating $m = 0.7$ and weighting $\eta = 0.5$, and together these values behave such that smaller settings recover a standard distillation baseline while larger settings amplify the inductive bias of our approach. In addition, several minor scalar ratios $\lambda_{\mathrm{KD}}$ and $\lambda_{\mathrm{CE}}$. All experiments use AdamW with batch size 32, maximum sequence length 128, weight decay 0.01, gradient clipping at 1.0, and 10% linear warmup with cosine decay.

| Dataset | Banking77 | Clinc150 | TREC | AGNews | Massive | Avg |
|---|---|---|---|---|---|---|
| FT Teacher | 93.70 | 96.35 | 97.40 | 94.69 | 89.47 | 94.32 |
| **Student (6L-768D)** | | | | | | |
| TinyBERT[†] | 89.47 $_{\pm 0.50}$ | 88.43 $_{\pm 0.06}$ | 96.13 $_{\pm 0.12}$ | 94.18 $_{\pm 0.15}$ | 88.84 $_{\pm 0.40}$ | 91.41 |
| MGSKD[†] | 91.58 $_{\pm 0.04}$ | 95.42 $_{\pm 0.05}$ | 87.60 $_{\pm 0.20}$ | 94.64 $_{\pm 0.15}$ | 88.56 $_{\pm 0.32}$ | 91.56 |
| PKD | 89.87 $_{\pm 0.15}$ | 95.09 $_{\pm 0.17}$ | 96.40 $_{\pm 0.20}$ | 94.54 $_{\pm 0.24}$ | 88.17 $_{\pm 0.20}$ | 92.81 |
| LKD | 93.03 $_{\pm 0.11}$ | 93.51 $_{\pm 0.44}$ | 96.07 $_{\pm 0.23}$ | 94.23 $_{\pm 0.29}$ | 88.94 $_{\pm 0.21}$ | 93.33 |
| AD-KD | 92.66 $_{\pm 0.45}$ | 95.42 $_{\pm 0.37}$ | 96.00 $_{\pm 0.35}$ | 94.57 $_{\pm 0.31}$ | 89.30 $_{\pm 0.28}$ | 93.59 |
| CKD | 93.54 $_{\pm 0.24}$ | 95.93 $_{\pm 0.17}$ | **97.53** $_{\pm 0.46}$ | 94.72 $_{\pm 0.15}$ | 89.30 $_{\pm 0.25}$ | 94.20 |
| **CUD (Ours)** | **93.83** $_{\pm 0.06}$ | **96.07** $_{\pm 0.14}$ | 97.40 $_{\pm 0.35}$ | **94.91** $_{\pm 0.12}$ | **89.44** $_{\pm 0.06}$ | **94.33** |
| **Student (4L-256D)** | | | | | | |
| LKD | 80.79 $_{\pm 0.44}$ | 88.31 $_{\pm 0.05}$ | 95.80 $_{\pm 0.00}$ | 94.28 $_{\pm 0.04}$ | 82.00 $_{\pm 0.04}$ | 88.24 |
| AD-KD | 82.22 $_{\pm 0.57}$ | 91.35 $_{\pm 0.08}$ | 95.06 $_{\pm 0.12}$ | 94.03 $_{\pm 0.05}$ | 84.17 $_{\pm 0.47}$ | 89.37 |
| PKD | 86.60 $_{\pm 0.39}$ | 93.27 $_{\pm 0.15}$ | 95.80 $_{\pm 0.53}$ | **94.37** $_{\pm 0.13}$ | **87.23** $_{\pm 0.07}$ | 91.45 |
| CKD | 89.86 $_{\pm 0.19}$ | 91.70 $_{\pm 0.15}$ | 95.40 $_{\pm 0.35}$ | 94.24 $_{\pm 0.09}$ | 86.60 $_{\pm 0.22}$ | 91.56 |
| **CUD (Ours)** | **91.22** $_{\pm 0.02}$ | **93.45** $_{\pm 0.05}$ | **96.47** $_{\pm 0.23}$ | 94.11 $_{\pm 0.27}$ | 87.22 $_{\pm 0.07}$ | **92.49** |

[†] Methods are trained with distillation already at the pretraining stage; we fine-tune publicly available checkpoints.

Table 1: **Main experimental results.** Accuracy on five classification benchmarks for two student sizes (4L-256D and 6L-768D) distilled from a fine-tuned teacher. The best performance is highlighted in **bold**, and the second-best is underlined. Our proposed CUD consistently outperforms prior KD approaches across datasets, with especially pronounced gains on high-cardinality tasks and for the smaller student, highlighting its effectiveness under challenging transfer regimes.

**Benchmarks & Evaluation.** We evaluate our method on five classification datasets spanning a wide range of label granularities, from 4 to 150 classes: Banking77 (77 intents) (Larson et al., 2019), CLINC150 (150 intents) (Casanueva et al., 2020), MASSIVE (60 intents) (FitzGerald et al., 2022), TREC (6 question types) (Li & Roth, 2002), and AG News (4 topic categories) (Zhang et al., 2015). Detailed dataset statistics are provided in Table 6 of the Appendix C. Additional results on binary classification settings are also included in the Appendix for completeness.

**Baselines.** We compare against representative KD methods, summarized by their characteristics:

1. **LKD (Logit-KD)** (Hinton et al., 2015): Response-based KL matching on teacher soft logits (with temperature). Simple and task-agnostic but ignores intermediate structure.

2. **PKD (Patient KD)** (Sun et al., 2019): Layer-wise alignment of hidden representations. *Exploits intermediate features and depth supervision*; effective when student width/layers can be mapped to the teacher, less suited to severe architectural mismatch.

3. **TinyBERT** (Jiao et al., 2020): Distills both self-attention and hidden states, originally with two-stage training and data augmentation. *Achieves high compression quality via rich feature-level objectives*; in our setup, we disable augmentation for a fair task-specific comparison.

4. **CKD (Contextual KD)** (Park et al., 2021): Transfers *relational knowledge* by modeling two forms of contextual structure: *Word Relation* between tokens, and *Layer Transforming Relation* across layers. Architecture-agnostic and effective for heterogeneous teacher–student pairs.

5. **MGSKD (Multi-Granularity Structural KD)** (Liu et al., 2022): Distills *structural relations* over *multiple semantic granularities* (tokens, spans, samples), formulated via pair-wise interactions and triplet geometric angles. Preserves richer linguistic structure than mono-granularity KD, at the cost of higher relational computation.

6. **AD-KD** (Wu et al., 2023): Response-based distillation with confidence/uncertainty-aware reweighting or calibration. *Mitigates propagation of over-confident teacher errors* while keeping architecture independence and low overhead.

## 4.2 MAIN RESULTS

Table 1 reports both teacher side and student side performance. Across all datasets and student capacities, CUD consistently outperforms prior baselines.

| Clinc150 | AUROC | FPR95 | FPR90 |
|---|---|---|---|
| TinyBERT | 79.06 | 70.04 | 54.55 |
| CKD | 92.21 | 38.97 | 22.53 |
| AD-KD | 93.91 | 30.08 | 13.31 |
| CUD (Ours) | **94.52** | **24.24** | **12.77** |

| Massive | AUROC | FPR95 | FPR90 |
|---|---|---|---|
| TinyBERT | 80.39 | 68.63 | 53.17 |
| CKD | 93.74 | 37.45 | 19.37 |
| AD-KD | 95.80 | 29.59 | 10.99 |
| CUD (Ours) | **96.79** | **20.12** | **8.90** |

| AGNews | AUROC | FPR95 | FPR90 |
|---|---|---|---|
| TinyBERT | 85.28 | 58.47 | 41.93 |
| CKD | 95.66 | 27.50 | 11.86 |
| AD-KD | 98.41 | 8.98 | 2.02 |
| CUD (Ours) | **99.07** | **4.53** | **1.56** |

| TREC | AUROC | FPR95 | FPR90 |
|---|---|---|---|
| TinyBERT | 82.34 | 61.80 | 42.60 |
| CKD | 95.18 | 31.40 | 15.00 |
| ADKD | 97.30 | 19.20 | 7.20 |
| CUD (Ours) | **97.45** | **16.60** | **8.60** |

Table 2: **OOD uncertainty performance across datasets.** The teacher model is fine-tuned on the in-distribution Banking77 dataset, whereas evaluations are conducted on each target test distribution.

| Task | TinyBERT | CKD | AD-KD | CUD (Ours) |
|---|---|---|---|---|
| Banking77 | 0.858 / 1.696 | 0.747 / 1.481 | 0.791 / 1.557 | 0.277 / 1.033 |
| Clinc150 | 0.861 / 1.741 | 0.835 / 1.657 | 0.730 / 1.480 | 0.236 / 1.023 |
| TREC | 0.929 / 1.851 | 0.870 / 1.675 | 0.950 / 1.900 | 0.566 / 0.993 |
| MASSIVE | 0.692 / 1.469 | 0.764 / 1.540 | 0.833 / 1.667 | 0.586 / 1.213 |
| AGNews | 0.794 / 1.393 | 0.972 / 1.919 | 0.865 / 1.570 | 0.801 / 1.396 |

Table 3: $ECE_{wrong}$ / $Brier_{wrong}$ calibration performance across tasks on wrong predictions

**Effect of student capacity.** For the larger student (6L 768D), several baselines already achieve accuracy comparable to the teacher (teacher average 94.32). In this regime CUD still attains the best average student performance (94.33), improving over vanilla logit KD by about $+1.0$ point on average (94.33 versus 93.33). For the smaller student (4L 256D), the dimensionality gap to the teacher makes representation–level matching difficult or even infeasible, so performance is largely determined by the quality of the response targets. In this regime, CUD delivers substantially clearly outperforming other baselines with larger margin.

**Effect of the cardinality.** CUD yields much larger gains, especially on tasks with many classes such as Banking77, CLINC150, MASSIVE, where it improves over vanilla KD by $+10.43$, $+5.14$, and $+5.22$ points respectively. This pattern matches our intuition: on tasks with many classes, the calibrated targets keep meaningful probability on several plausible labels while suppressing the dominant wrong peak, so the student can imitate a richer set of alternatives. In contrast, when the number of classes is small, for example (AGNews, TREC), the teacher distribution is already almost deterministic and offers little additional structure to recover. For extremely small numbers of classes (binary), this effect becomes stronger; suppressing a wrong peak acts like mild logit re-tempering and adds little information beyond standard smoothing or temperature scaling. Detailed analysis are provided in the Appendix D.

## 4.3 UNCERTAINY & OOD ROBUSTNESS

Our approach is explicitly designed to shape predictive uncertainty, so we expect it to provide more reliable confidence estimates and stronger performance on Out-of-Distribution (OOD) detection than standard KD baselines. We therefore evaluate both OOD detection and calibration quality.

**OOD Detection** We follow a standard OOD detection protocol: the teacher and student are trained on Banking77 (in distribution - IND), and we treat CLINC150, MASSIVE, and other intent datasets with different domains and label spaces as OOD. We compute AUROC, FPR95, and FPR90 to quantify how well IND and OOD samples are separated. As shown in Table 2, CUD consistently achieves the highest AUROC and the lowest FPR across all OOD benchmarks. The gains are particularly large on heterogeneous OOD datasets such as CLINC150 and MASSIVE, even though the teacher and

| Configuration | Banking77 | CLINC150 | TREC | AGNews | MASSIVE | Avg | AVG Δ |
|---|---|---|---|---|---|---|---|
| **Teacher** (*FT, no KD*) | 93.70 | 96.35 | 97.40 | 94.69 | 89.47 | 94.32 | — |
| *Student (4L–256D) with distillation variants* | | | | | | | |
| LKD | 80.61 | 88.35 | 95.80 | **94.31** | 82.04 | 88.22 | |
| LKD + DUS | 90.62 | 93.24 | 96.40 | 94.17 | 86.76 | 92.24 | +4.02 |
| LKD + W-Clip | 89.25 | 93.02 | 95.40 | 94.28 | 86.37 | 91.66 | +3.44 |
| **CUD (Ours)** | **91.22** | **93.45** | **96.47** | 94.11 | **87.22** | **92.49** | **+4.27** |

Table 4: **Ablation of our model on multiclass benchmarks.** We evaluate the individual contributions of DUS and W-Clip in a 4-layer, 256-dim student. Both modules provide measurable improvements over LKD, and their combination in CUD delivers the strongest gains.

student never see these domains during training. This indicates that the uncertainty shaped by DUS transfers to stronger OOD rejection, not just higher in distribution accuracy.

**Uncertainty Estimation**    To evaluate how well each model estimates uncertainty, we compute Expected Calibration Error (ECE) and Brier score on the test set of each dataset. In Table 3, we report ECE computed only over wrong predictions, where DUS achieves markedly lower values than all KD baselines across every dataset. This shows that DUS is more capable of signaling its own mistakes, assigning lower confidence when it is wrong, while competing methods remain overconfident even on errors. Brier scores follow the same trend, confirming that DUS provides more informative uncertainty estimates; detailed results are included in Tab. 8 and Fig. 3. We also assess whether confidence separates correct from incorrect predictions by plotting ROC curves on BANKING77 (in distribution test, label equals correctness). As shown in Figure 2a, our method attains a higher AUROC than vanilla KD (0.92 vs. 0.83), with a visibly steeper rise in the low FPR region, where selective prediction operates. This shows that calibrated targets from CUD produce confidence scores that better rank easy and hard examples, yielding stronger risk coverage trade offs. Ablations further reveal that DUS and W Clip individually improve correctness AUROC over vanilla KD (0.90 and 0.88 vs. 0.83), while their combination in CUD yields the strongest separation (0.92), suggesting complementary effects.

## 5 ANALYSIS

### 5.1 ABLATION STUDY

Table 4 reports ablations of our main modules (DUS and W-Clip) in the 4L–256D student. While both components improve over vanilla KD on average, and the gains grow with label cardinality as noted in the main results, the detailed breakdown further reveals when and why they matter.

**Closing the teacher–student gap.**    Relative to the fine-tuned teacher, vanilla KD leaves substantial headroom: $-13.09$ points on Banking77 (80.61 vs. 93.70), $-8.0$ on CLINC150 (88.35 vs. 96.35), and $-7.43$ on MASSIVE (82.04 vs. 89.47).CUD closes 80%, 63%, and 75% of these gaps respectively, demonstrating that calibrated uncertainty specifically helps students recover performance where naive KD collapses.

**Relative impact of modules.**    DUS alone outperforms W-Clip in average accuracy (92.24 vs. 91.66), showing that reshaping the teacher distribution is more fundamental than just reweighting samples. W-Clip is beneficial in high-class regimes (e.g., $+8.64$ on Banking77 and $+4.33$ on MASSIVE vs. vanilla KD), but can over-emphasize noise in easier tasks. The full CUD system balances both: projection (R2) prevents over-amplification of wrong peaks while clipped weighting emphasizes informative errors, producing the most stable and highest average (92.49, $\Delta+4.27$).

**Teacher calibration.**    In our setup, DUS trained teacher attains slightly lower accuracy than a standard fine tuned teacher (Tab. 11), yet students distilled from it consistently outperform those distilled from the standard teacher. This holds not only for our CUD objective but also for other KD baselines when we simply replace their original teacher with ours (teacher side ablations in Tab. 10).

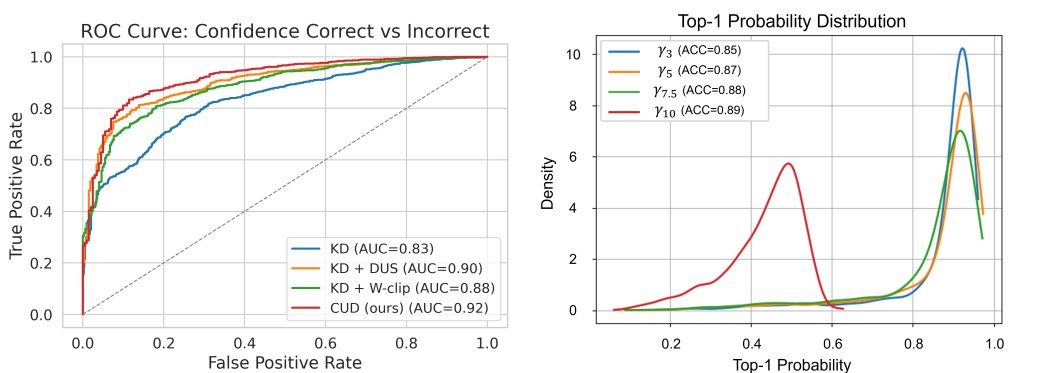

(a) ROC curves comparing confidence-based correctness separation.

(b) Teacher probability distributions under varying $\gamma$. Higher $\gamma$ values soften over-confident peaks.

Figure 2: Effect of calibration on uncertainty and distributions.

## 5.2 THE ROLE OF $\gamma$: BALANCING ACCURACY AND CALIBRATED UNCERTAINTY

We study how $\gamma$ in DUS reshapes the teacher distribution and student performance. Since the per-example weight scales as $(1 - p_T^y)^\gamma$, increasing $\gamma$ aggressively down-weights correct examples and emphasizes wrong ones. Empirically (Fig. 2b), larger $\gamma$ produces a *softer* top-1 distribution: the sharp peak near $\approx 0.95$ collapses while probability mass shifts toward mid confidence (0.5–0.9). Thus, there is a clear **inverse relation** between $\gamma$ and top-1 concentration/peak height (and, equivalently, a positive relation with entropy on hard cases). This softening is beneficial for transfer up to a point. On the dev sweep shown, student accuracy rises monotonically as $\gamma$ increases from 3 to 10 (0.85→0.89); table at right), indicating that *within this range* recovering teacher uncertainty improves the ranking signal distilled to the student. In our experiments, the $\gamma$ sweep further illustrates the transfer-first trade-off: modest increases in teacher uncertainty (sometimes with a minor drop in the teacher's standalone accuracy) correlate with *higher* student accuracy after KD, reinforcing that calibrating the teacher for *distillability* can be preferable to maximizing its peak accuracy.

## 6 RELATED WORK

Knowledge distillation (KD) fundamentally relies on transferring the "knowledge" encoded in soft targets (Hinton et al., 2015), yet standard teachers often exhibit overconfidence that degrades this signal. While calibration techniques (Guo et al., 2017) and regularizers like label smoothing (Müller et al., 2019) or confidence penalties (Pereyra et al., 2017) address predictive sharpness, they risk indiscriminately blurring the fine-grained inter-class geometry essential for effective transfer. Furthermore, prior attempts to optimize teachers for distillation (Menon et al., 2021; Dong et al., 2024) or align intermediate features (Sun et al., 2019) often lack difficulty-awareness or impose strict architectural constraints. In contrast, our approach targets the *distributional shape* of the transfer; we jointly optimize the teacher to maintain difficulty-aware uncertainty through selective entropy shaping (DUS) and apply a constrained projection (W-Clip) to suppress specific wrong-class peaks without distorting the informative pairwise relations among background classes. For an extended review of related literature, please refer to the Appendix J.

## 7 CONCLUSION

We presented *Calibrated Uncertainty Distillation (CUD)*, a simple architecture-agnostic dual distillation mechanism. Unlike conventional approaches that treat the teacher as fixed, CUD explicitly reshapes the teacher to retain difficulty-aware uncertainty and calibrates the transfer to suppress over-confident mistakes while preserving informative class relations. Empirically, CUD delivers consistent surpass stronger other baselines, highlighting the value of calibrated distributions over sharpened certainty.

## LIMITATIONS

Our work has several limitations. First, we restrict evaluation to single-label classification, leaving open the extension to multi-label, sequence-level, or generative tasks where calibration constraints may require new formulations. Second, our projection rules rely on fixed hyperparameters (e.g., entropy weights, wrong-mass budget), which, while effective, may not optimally adapt across domains or dynamic training conditions. Third, the benefits of calibrated uncertainty are reduced on binary or very small-class problems, where the teacher's uncertainty structure becomes nearly one-dimensional; in such cases, suppressing a wrong peak is almost equivalent to mild logit re-tempering, providing limited additional signal beyond classical smoothing or temperature scaling. Finally, our experiments are conducted on standard benchmarks; validation under real-world distribution shift, selective prediction, and resource-limited scenarios is an important next step. These limitations suggest several directions for future work: adaptive per-example calibration, integration with semantic neighborhoods or taxonomies, and applications beyond classification toward generative modeling. We believe exploring these avenues will further enhance the role of calibrated uncertainty in distillation and broaden the impact of compact, trustworthy models in practice.

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

## A  Theoretical Analysis of the Constrained Projection

In this appendix, we provide the rigorous mathematical justification for the projection problem defined in Section 2, addressing the existence and uniqueness of the solution and deriving the connection between the optimal solution and our proposed heuristics.

### A.1  Existence and Uniqueness of the Solution

The optimization problem posed in Eq. (2) is to find a distribution $q$ that minimizes the Kullback-Leibler (KL) divergence $D_{KL}(q\|p_T)$ subject to linear inequality constraints defined by R1 and R2.

**Theorem 1 (Uniqueness of Projection)** *Let $\mathcal{Q}$ be the set of valid probability distributions on the simplex $\Delta^K$. The feasible set $\mathcal{C} = \{q \in \mathcal{Q} \mid R1(q) \leq \epsilon_1, R2(q) \leq \epsilon_2\}$ is defined by the intersection of linear half-spaces and the probability simplex, making $\mathcal{C}$ a convex polytope. Since the objective function $f(q) = D_{KL}(q\|p_T)$ is strictly convex with respect to $q$ (for $q, p_T > 0$), and the feasible set $\mathcal{C}$ is convex and compact, standard convex optimization theory guarantees that if $\mathcal{C}$ is non-empty, there exists a **unique** global minimizer $q^*$.*

Above guarantees that our target calibration objective does not suffer from instability due to multiple local minima.

### A.2  KKT Analysis and Exponential Tilting

To characterize the solution, we formulate the Lagrangian $\mathcal{L}$. Let constraints R1 and R2 be written generally as linear constraints $\sum_{k \in S} q_k \leq C$. Introducing Lagrange multipliers $\mu$ for the neighborhood lower-bound constraint (R1) and $\nu$ for the wrong-mass upper-bound constraint (R2), the KKT stationarity condition yields:

$$\frac{\partial \mathcal{L}}{\partial q_k} = \log q_k - \log p_T(k) + 1 + \sum \lambda_i \frac{\partial \text{Constraints}}{\partial q_k} = 0 \tag{10}$$

Solving for $q_k$, we obtain the closed-form solution:

$$q_k^* = \frac{1}{Z} p_T(k) \exp\left(\mu \cdot \mathbb{I}(k \in N) - \nu \cdot \mathbb{I}(k \in W)\right) \tag{11}$$

where $Z$ is the normalization constant. This confirms that the optimal projection is an *exponential tilting* of the teacher distribution, scaling the probabilities in the constrained sets $N$ and $W$ while preserving the relative ratios of probabilities for unconstrained classes (the "dark knowledge").

### A.3  W-Clip as a First-Order Approximation

We now show that the heuristic W-Clip operation approximates the optimal solution derived in Eq. (11). Consider the case where the wrong-mass constraint is active for the top-1 class $k^*$, i.e., $W = \{k^*\}$. The optimal update requires scaling $p_T(k^*)$ by a factor $e^{-\nu}$.

Performing a first-order Taylor expansion around $\nu = 0$ (assuming a conservative constraint violation):

$$e^{-\nu} \approx 1 - \nu \tag{12}$$

Substituting this into the update rule:

$$q_{k^*}^* \approx \frac{1}{Z} p_T(k^*)(1 - \nu) \approx p_T(k^*) - \nu p_T(k^*) \tag{13}$$

This linear subtraction form is functionally equivalent to the W-Clip operation defined in Eq. (8), where the clipped mass $\eta$ corresponds to the term $\nu p_T(k^*)$. Thus, W-Clip provides a computationally efficient, first-order approximation to the theoretically optimal exponential projection, avoiding the need for iterative solvers during the training loop.

## B  HYPERPARAMETER SENSITIVITY ANALYSIS

| $\lambda_H$ | $m$ | $\eta$ | $\lambda_{KD}$ / $\lambda_{CE}$ |
|---|---|---|---|
| **0.1 : 91.22** | 0.1 : 91.21 | 0.1 : 91.19 | 0.2 / 0.8 : 90.73 |
| 0.2 : 91.16 | 0.3 : 91.22 | 0.3 : 91.22 | 0.4 / 0.6 : 91.04 |
| 0.3 : 91.04 | 0.5 : 91.22 | **0.5 : 91.23** | 0.6 / 0.4 : 91.21 |
| 0.4 : 91.13 | **0.7 : 91.23** | 0.7 : 91.22 | **0.8 / 0.2 : 91.22** |
| — | 0.9 : 91.22 | 0.9 : 91.24 | — |

Table 5: **Hyperparameter sensitivity results.** Bold values indicate the configuration used in the main experiments.

To better understand the robustness of our hyperparameter choices, we conduct a sensitivity study across four key parameters. Bold values in each table correspond to the settings used for our main experimental results. As shown, CUD exhibits low sensitivity across a broad range of configurations, further justifying our decision to perform minimal per-dataset tuning.

## C  DETAILED BENCHMARK DESCRIPTION

| Dataset | #Classes | Domain | Task Type | Description |
|---|---|---|---|---|
| **Banking77** | 77 | Customer Support | Intent Classification | A fine-grained intent dataset for banking-related customer queries (Larson et al., 2019). |
| **CLINC150** | 150 | Open-domain | Intent Classification | Wide-coverage natural language intent dataset with 150 classes across multiple domains (Casanueva et al., 2020). |
| **MASSIVE** | 60 | Multilingual SLU | Intent Classification | Multilingual dataset for SLU tasks; English subset used unless otherwise stated (FitzGerald et al., 2022). |
| **TREC** | 6 | Question Answering | Question Type Classification | Coarse-grained question type labels for factoid QA tasks (Li & Roth, 2002). |
| **AG News** | 4 | News | Topic Classification | News article classification into 4 broad topical categories (Zhang et al., 2015). |

Table 6: Metadata summary of the benchmark datasets used in our evaluation..

We provide here a metadata summary of all benchmark datasets used in our experiments. This table outlines class cardinalities, domains, and task types, offering additional context for the diversity of evaluation settings.

## D  ON BINARY CLASSIFICATION

| Task | FT Teacher | LKD | PKD | TinyBERT[†] | AD-KD | CUD (Ours) |
|---|---|---|---|---|---|---|
| QNLI | 90.66 | 89.09 | 89.65 | **90.50** | 89.84 | 89.23 |
| SST-2 | 90.82 | 90.71 | 90.82 | 91.60 | **91.62** | 91.28 |

[†] Methods are trained with distillation already at the pretraining stage; we fine-tune publicly available checkpoints.

Table 7: **Binary-task comparison on QNLI and SST-2** in a 6-layer, 768-dim student.

As discussed in the main text, calibrated uncertainty distillation is most effective when the teacher distribution contains rich multi-class structure, which provides informative relative probabilities over

many plausible alternatives. In contrast, for binary or near-binary tasks, the teacher's output reduces to a one-dimensional distribution of the form $(p, 1 - p)$, limiting the amount of dark knowledge available for transfer.

To verify this empirically, we report results on two standard binary GLUE tasks, QNLI and SST-2. Table 7 compares Vanilla KD, DUS, W-Clip, and the full CUD across student architectures. Consistent with our analysis, the gains from calibrated uncertainty are substantially smaller than on high-cardinality tasks (e.g., Banking77 or CLINC150). In some cases, the performance is nearly identical to Vanilla KD, reflecting that suppressing a wrong peak in the binary case is almost equivalent to applying a temperature rescaling to the teacher logits.

Overall, these findings reinforce a key property of calibrated uncertainty distillation: its advantages grow with class cardinality and class ambiguity. In binary tasks—where the teacher's uncertainty structure collapses to a single degree of freedom—the potential benefit of redistributing probability mass is inherently limited. Nevertheless, our method remains competitive and does not degrade accuracy, indicating that CUD is a safe replacement for conventional KD even in low-cardinality settings.

## E    ADDITIONAL RESULTS ON UNCERTAINTY QUANTIFICATION

| Task | TinyBERT | CKD | AD-KD | CUD (Ours) |
|------|----------|-----|-------|------------|
| Banking77 | 0.076 / 0.203 | 0.047 / 0.159 | 0.039 / 0.112 | 0.359 / 0.237 |
| Clinc150 | 0.066 / 0.203 | 0.028 / 0.067 | 0.021 / 0.068 | 0.394 / 0.228 |
| TREC | 0.013 / 0.061 | 0.023 / 0.048 | 0.021 / 0.047 | 0.182 / 0.099 |
| MASSIVE | 0.059 / 0.188 | 0.080 / 0.223 | 0.074 / 0.182 | 0.101 / 0.176 |
| AGNews | 0.009 / 0.095 | 0.047 / 0.099 | 0.027 / 0.086 | 0.061 / 0.095 |

Table 8: ECE / Brier calibration performance across tasks.

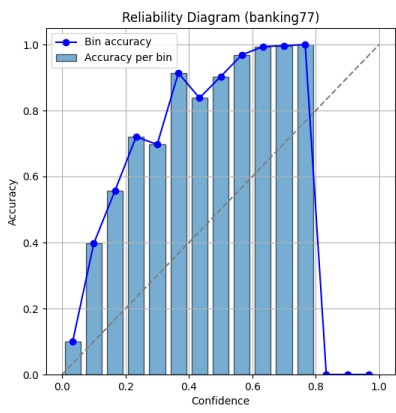
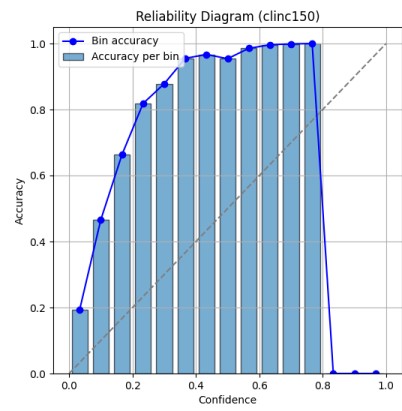

(a) Reliability diagram on Banking77.                         (b) Reliability diagram on Clinc150.

Figure 3: **Effect of DUS on predictive calibration.** Across datasets, DUS reduces overconfident predictions by reshaping the teacher distribution, leading to a controlled underconfident shift. Although this increases ECE due to lower confidence than accuracy, the behavior is safer than over-confidence and yields more reliable uncertainty estimates for downstream OOD detection and risk-sensitive settings.

Table 8 reports the Brier and ECE scores computed over all predictions, and Figure 3 shows the plot of the confidence.

| Method | Banking77 | Clinc150 | Massive | TREC | AGNews | Avg |
|---|---|---|---|---|---|---|
| Temperature Scaling (TS) | 80.61 | 88.35 | 82.04 | 95.80 | 94.31 | 88.22 |
| Label Smoothing (LS) | 81.29 | 86.88 | 82.19 | 95.80 | 94.19 | 88.07 |
| CUD (Ours) | 91.23 | 93.43 | 87.21 | 96.20 | 94.00 | 92.41 |

Table 9: **Comparison of smoothing strategies.** Temperature Scaling (TS) and Label Smoothing (LS) rely on simple, rule-based smoothing, whereas our DUS-based method reshapes the teacher's scores in a way that is more suitable for knowledge distillation.

## F COMPARISON WITH TS AND LS.

We also compare our calibrated distillation against simpler smoothing baselines such as Temperature Scaling (TS) and Label Smoothing (LS). This comparison highlights that merely smoothing the teacher's distribution—without modeling sample-dependent structure—does not preserve the rich information needed for effective distillation. Table 9 reports a direct comparison between TS, LS, and our approach under the same experimental setup.

All three methods are evaluated under an identical teacher–student architecture. The student is a 4-layer, 256-dimensional BERT model (`google/bert_uncased_L-4_H-256_A-4`). For each method, we perform a grid search over learning rates $\{5 \times 10^{-4}, 10^{-4}, 5 \times 10^{-5}\}$. All other training settings (optimizer, batch size, number of epochs, etc.) are kept fixed so that the only difference lies in how the teacher's distribution is smoothed.

As shown in Table 9, our method consistently improves over TS and LS in most cases. The gains are especially notable on high-cardinality tasks such as Banking77, CLINC150, and MASSIVE. This suggests that, compared to simple rule-based methods that apply a uniform temperature or fixed label smoothing, our loss preserves richer dark knowledge by adapting the teacher distribution to each sample's difficulty and uncertainty. In other words, our approach does more than merely "soften" the teacher: it reshapes the teacher's confidence structure in a way that is more conducive to effective knowledge distillation.

## G ADDITIONAL TEACHER SIDE ABLATIONS

| Baseline | Method | Banking77 | Clinc150 | TREC | AGNews | Massive |
|---|---|---|---|---|---|---|
| TinyBERT | ft teacher | 89.74 | 88.44 | 96.20 | 94.31 | 89.12 |
|  | ours | **91.29** | **92.91** | **96.80** | **94.36** | **89.27** |
| MGSKD | ft teacher | 91.62 | 95.46 | 87.80 | **94.81** | 88.88 |
|  | ours | **91.92** | **95.91** | **97.20** | 94.44 | **89.27** |
| CKD | ft teacher | 93.76 | 96.11 | 97.80 | **94.89** | 89.57 |
|  | ours | **93.86** | **97.03** | 97.80 | 94.73 | **89.86** |
| AD-KD | ft teacher | 93.05 | 95.53 | 96.40 | **94.82** | **89.62** |
|  | ours | **93.08** | **95.77** | **97.60** | 94.55 | 89.32 |

Table 10: Comparison of layer-6 student performance using a fine-tuned teacher vs. our CUD-trained teacher.

| Method | Banking77 | Clinc150 | TREC | AGNews | MASSIVE |
|---|---|---|---|---|---|
| ft teacher | **93.70** | **96.35** | **97.40** | 94.69 | **89.47** |
| ours teacher ($\gamma$=5) | 91.62 | 94.17 | **97.40** | 94.42 | 89.12 |
| ours teacher ($\gamma$=10) | 92.43 | 94.93 | 97.00 | **94.72** | **89.47** |

Table 11: Teacher performance under different levels of DUS smoothing. The *ft teacher* row reports the accuracy of a standard fine-tuned teacher, while *ours teacher* corresponds to the same architecture trained with DUS at different smoothing strengths $\gamma$.

For KD baselines, we also evaluate the effect of replacing the conventional fine-tuned teacher with our CUD-trained teacher while keeping the student architecture and loss fixed. As shown in Table 10, almost all baselines and benchmarks exhibit consistent performance gains, indicating that substituting the standard teacher with our calibrated teacher provides clear benefits.

This suggests that, although the proposed teacher training procedure does not necessarily improve (and can even slightly reduce) the teacher's raw accuracy (cf. Table 11), it yields a teacher that encodes richer dark knowledge about inter-class relationships. In other words, even under the same distillation architecture, the quality of information transferred to the student depends strongly on how the teacher itself is trained, and our teacher provides more informative soft targets.

The effect is particularly pronounced on datasets with a large number of classes, such as Banking77, CLINC150, and MASSIVE. In these multi-class settings, decision boundaries and error patterns become more complex, so the teacher's ability to capture subtle relationships among classes has a direct impact on distillation performance. Our teacher better models the underlying uncertainty and correlations over this complex label space, thereby increasing the amount of useful information available to the student through soft targets.

By contrast, on datasets with relatively few classes, such as TREC and AGNews, the amount of exploitable dark knowledge is inherently limited. Consequently, the absolute gains from emphasizing this structure are smaller. Nevertheless, our teacher still matches or slightly surpasses the standard fine-tuned teacher on most of these tasks. Taken together, these results indicate that the proposed teacher training strategy yields substantial benefits in challenging multi-class regimes, while remaining a safe and stable improvement that does not degrade performance even in simpler, low-cardinality settings.

## H   COMPUTATION ANALYSIS

| Method | Wall-clock Time | Relative Cost |
|--------|-----------------|---------------|
| Standard Fine-tuning (FT Teacher) | 11m 06s (665.8 s) | $1.00\times$ |
| DUS Teacher (Ours) | 12m 10s (729.8 s) | $1.05\times$ |

Table 12: **Computation cost comparison during teacher training.** DUS incurs only a small overhead of approximately 5% relative to standard fine-tuning.

DUS introduces minimal computational overhead because it modifies only the loss function while leaving the model architecture and training pipeline unchanged. Theoretical FLOPs per iteration are therefore identical to those of standard fine-tuning.

It is important to note that our method—like the standard practice in response-based KD (e.g., PKD, TinyBERT, CKD, AD-KD, MGSKD)—assumes a task-specific fine-tuned teacher. Although the cost of obtaining such a teacher may appear "hidden" in real-world scenarios, where many pre-tuned large models are publicly available, this requirement is shared across nearly all KD pipelines. Under the assumption that a task-specific fine-tuned teacher—one that is not publicly available—must be trained for each task, the only additional computation introduced by DUS arises from the modified loss.

Table 12 reports the wall-clock time on BANKING77: DUS teacher training takes 12m 10s, compared to 11m 6s for standard fine-tuning, corresponding to an overhead of only 5%. The student-side distillation cost remains essentially unchanged, as the complexity of our loss is comparable to that of standard KL-based KD objectives.

Overall, DUS provides a meaningful perspective on knowledge distillation by reshaping the teacher into a more calibrated and informative distribution, while adding only negligible computational cost, even when accounting for the teacher fine-tuning step.

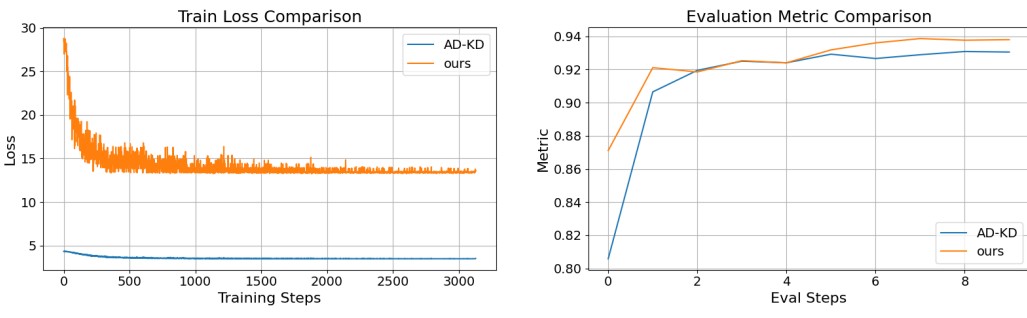

(a) Training loss curves.    (b) Validation accuracy curves across evaluation steps.

Figure 4: Learning curves illustrating the convergence and generalization behavior of our method. Our approach demonstrates smoother optimization and stronger validation performance throughout training.

# I    LEARNING CURVE

Figure 4 reports the learning curves comparing our method. The training loss of our method decreases smoothly and stabilizes early, demonstrating that the optimization process is well-behaved even though its absolute loss scale differs from AD-KD due to the different loss terms used.

Validation accuracy improves steadily and remains consistently higher than that of AD-KD throughout training, reaching a stable plateau without late-stage degradation.

Overall, these learning curves confirm that (1) our optimization process converges reliably, and (2) the performance gains are attributable to a better-shaped training objective rather than incidental optimization effects.

# J    EXTENDED RELATED WORK

**KD under uncertainty, calibration, and class structure.**    Classical response-based KD transfers a teacher's softened output distribution to the student (Hinton et al., 2015), leveraging "dark knowledge''—relative probabilities over non-ground-truth classes—which is especially informative when class cardinality is large. A parallel literature studies calibration and uncertainty of modern networks (Guo et al., 2017), and regularizers that alter predictive sharpness, such as label smoothing (Müller et al., 2019) and confidence penalties (Pereyra et al., 2017). While these techniques can improve expected calibration, they may also blur meaningful inter-class geometry that KD intends to pass on. Self-/born-again distillation variants (Mobahi et al., 2020; Zhang et al., 2019) improve students without an external teacher, but they do not explicitly preserve difficulty-aware uncertainty or prevent wrong-peak propagation. Our approach targets the *distributional shape* transferred by KD: we (i) raise teacher entropy selectively on hard inputs to keep informative neighborhoods (R1), and (ii) project the teacher outputs to respect a wrong-mass budget (R2), thus preserving pairwise odds for the untouched classes.

**Teacher shaping for distillation.**    Beyond treating the teacher as fixed, several works revisit how teachers should be trained for better students. Menon et al. (2021) argue that soft targets closer to the latent label distribution yield stronger generalization than one-hot supervision; Dong et al. (2024) further reformulate teacher training with regularizers that move predictions toward a faithful class distribution for the data. Orthogonally, focal loss (Lin et al., 2017) down-weights trivially correct examples and emphasizes hard ones; we adapt this idea to build a *distillation-friendly* teacher by coupling a focal term with a gated entropy reward, thereby increasing uncertainty *only* where the teacher is wrong or uncertain, unlike global smoothing. Orthogonally, focal loss down-weights trivially correct examples and emphasizes hard ones. Recent work by (Hamidi et al., 2024) also reformulates teacher training to better align predictions with the latent data distribution. While sharing the goal of optimizing teachers, our approach specifically targets informational calibration rather than

distributional matching; we couple a focal term with a gated entropy reward to build a *distillation-friendly* teacher that increases uncertainty *only* where it is epistemic, unlike global smoothing or pure accuracy-driven shaping.

**Selective imitation and error-aware KD.**    Confidence-aware distillation methods reweight or filter teacher supervision based on teacher confidence, aiming to reduce the harm of over-confident mistakes while keeping the simplicity of response-based KD (e.g., "adaptive'' or "selective'' KD variants). Feature-based approaches such as PKD (Sun et al., 2019) and TinyBERT (Jiao et al., 2020) align intermediate representations (hidden states, attention) and can be very effective when student width/depth can be mapped to the teacher; however, they are less applicable under severe architectural mismatch (e.g., depth *and* width changes). Our method is architecture-agnostic: rather than aligning features, we enforce a small *projection* on the teacher distribution that (a) suppresses only the wrong top-1 mass under a budget (R2) and (b) leaves the rest of the class relations intact, which theory and our results indicate is most beneficial as the number of classes grows.

Relative to prior KD, our contribution is to *jointly* (1) shape the teacher to preserve difficulty-aware uncertainty and (2) calibrate the transfer via a constrained projection that prevents error propagation. This coupling operationalizes the intuition that KD helps most when the teacher behaves like a calibrated posterior on both easy and hard inputs, and explains the empirical trend that our gains increase with class cardinality while becoming comparable to temperature scaling or smoothing on binary tasks.

# K ALGORITHM

---

**Algorithm 1** Proposed Knowledge Distillation Procedure

---

**Require:** Teacher $T$, Student $S$

  **Step 1: Teacher Training**
  **for** epoch **do**
     $T(x) \leftarrow$ teacher logits
     $\mathcal{L}_{CE} \leftarrow$ CrossEntropy$(T(x), y)$
     $\mathcal{L}_{focal} \leftarrow$ focal loss on $(T(x), y)$
     $\mathcal{L}_{ent} \leftarrow$ entropy loss on wrong samples
     $\mathcal{L}_{total} \leftarrow \mathcal{L}_{CE} + \mathcal{L}_{focal} + \mathcal{L}_{ent}$
     Update $T$ with $\mathcal{L}_{total}$
  **end for**

  **Step 2: Student Distillation**
  **for** epoch **do**
     $S(x) \leftarrow$ student logits
     **if** teacher correct **then**
       $T'(x) \leftarrow T(x)$
     **else**
       Post-process $T(x) \rightarrow T'(x)$
     **end if**
     $\mathcal{L}_{CE} \leftarrow$ CrossEntropy$(S(x), y)$
     $\mathcal{L}_{KD} \leftarrow$ KL$(S(x) \parallel T'(x))$
     $\mathcal{L}_{total} \leftarrow \mathcal{L}_{CE} + \mathcal{L}_{KD}$
     Update $S$ with $\mathcal{L}_{total}$
  **end for**

---

