# OpenReview forum: "TRUST THE UNCERTAIN TEACHER: DISTILLING DARK KNOWLEDGE VIA CALIBRATED UNCERTAINTY"
_ICLR.cc/2026/Conference — Submitted to ICLR 2026_

### Official Review · Reviewer_x5wW · 2025-10-29

**Soundness:** 3
**Presentation:** 3
**Contribution:** 3
**Rating:** 6
**Confidence:** 3

**Summary:**

This paper introduces Calibrated Uncertainty Distillation (CUD), which reformulates knowledge distillation as a two-stage process. In Stage 1, a Difficulty-aware Uncertainty Shaping (DUS) module is employed. It uses a gated entropy bonus combined with focal loss to increase predictive entropy, thereby redistributing probability mass toward semantically related non-maximal classes. In Stage 2, Wrong-mass Clipping (W-Clip) is proposed to address over-confident yet incorrect teacher predictions. It transfers the smaller value between a predefined budget and the margin between the top-1 incorrect class and the ground-truth class to the correct class, yielding a wrong-mass–budgeted soft target. Finally, the student is trained using a tempered KL divergence toward the calibrated teacher distribution, optionally combined with a cross-entropy loss.

**Strengths:**

1. The authors provide a clear and well-supported analysis showing that cross-entropy–trained teachers suffer from overconfidence and distribution collapse, which result in the loss of dark knowledge.
2. Both DUS and W-Clip are lightweight and architecture-agnostic modules that can be seamlessly plugged into existing systems, offering low implementation cost and high practicality.

**Weaknesses:**

1. The experiments are limited to text datasets, without validation on other modalities, which restricts the generality of the conclusions.
2. The compared baseline methods appear somewhat outdated.
3. Many knowledge distillation methods have been proposed to mitigate the over-confidence of teacher models. The authors should clarify what concrete advantages their approach offers, particularly considering that the comparisons and data modalities are relatively limited.
Moreover, since teacher models are usually large, fine-tuning them can incur considerable computational cost. It would be helpful to discuss how the training overhead of the proposed method compares to other approaches, and whether the increased cost is reasonable given the performance gains.

**Questions:**

Please see the Weaknesses.

---

> ### Author Response · Authors · 2025-11-22
>
> ### **Weakness 1: Regarding Generalization to Other Modalities**
>
> We fully agree that validating the proposed method across diverse modalities is a valuable direction. However, extending our evaluation to vision tasks is not directly transferable.
>
> The fundamental differences between modalities require non-trivial adaptations, so the baselines for vision tasks differ from those for NLP tasks. Meaning that our baselines cannot be applied to vision architectures without substantial modification.
>
> Instead, to address the core concern regarding model diversity and applicability to larger architectures, **we are actively conducting additional experiments using LLaMA.** We will share these results as soon as they become available.

---

> ### Author Response · Authors · 2025-11-22
>
> ### **Weakness 2: Regarding Outdated Baselines** - Section 4.1: Lines 310-316, Table 1
>
> We acknowledge the reviewer's concern that the initial baselines might appear outdated. To relieve this concern:
>
> 1. **Extension to Modern Architectures:** As detailed in our response previous comment, we fully agree that validating applicability to modern backbones is crucial. We are actively conducting additional experiments with **LLaMA** to verify our model’s transferrability. We will update those results as soon as they become available.
> 2. **Inclusion of Strong Baselines:** we have implemented and added two methods (**MGSKD** [1] and **CKD** [2] ) to our main performance table. The updated results demonstrate that CUD consistently outperforms these stronger baselines, validating its efficacy against more modern approaches.
>
> [1] Liu et al, Multi-Granularity Structural Knowledge Distillation for Language Understanding, ACL 2022
>
> [2] Park et al, Contextual Knowledge Distillation for Natural Language Understanding, AAAI 2021

---

> ### Author Response · Authors · 2025-11-22
>
> ### **Weakness 3 \& 4**
>
> **W3: Regarding Comparison with Calibration Baselines - Appendix F**
>
> We thank the reviewer for this insightful observation. In response, we conducted additional experiments comparing CUD against representative calibration methods (Label Smoothing - LS & Temperature Scaling - TS):
>
> **1. Conceptual Difference:**
>
> - **LS & TS (Global Smoothing):** These methods apply a global penalty that flattens the distribution uniformly across all samples. They cannot distinguish between *justified confidence* (on clear inputs) and *overconfidence* (on ambiguous inputs), often blurring the fine-grained class structure ("dark knowledge") essential for KD.
> - **CUD (Conditional Shaping):** In contrast, CUD does not merely "soften" logits post-hoc. It **reshapes the teacher during training** via a conditional gating mechanism. This acts as a smart filter: preserving sharpness on easy samples to maintain discriminative power, while selectively increasing entropy on hard samples to reflect true epistemic uncertainty.
>
> **2. Empirical Results**
>
> If global smoothing were sufficient for distillation, LS or TS should yield comparable gains to CUD. While LS and TS effectively mitigate numerical overconfidence, they fail to translate this into effective knowledge transfer. CUD’s **sample-adaptive mechanism** achieves a substantial improvement (**+4.19%p** on average), confirming that generating a *distillation-friendly* distribution requires dynamic shaping rather than blind smoothing.
>
> | **Method** | **Banking77(77)** | **Clinc150(150)** | **Massive(60)** | **TREC(6)** | **AGNews(4)** | **Avg** |
> | --- | --- | --- | --- | --- | --- | --- |
> | **TS** (Global) | 80.61 | 88.35 | 82.04 | 95.80 | **94.31** | 88.22 |
> | **LS** (Global) | 81.29 | 86.88 | 82.19 | 95.80 | 94.19 | 88.07 |
> | **CUD** (Ours) | **91.23** | **93.43** | **87.21** | **96.20** | 94.00 | **92.41** |
>
> **W4: Regarding Computational Cost vs. Justification - Appendix H**
>
> We acknowledge that teacher fine-tuning incurs computational costs, but we argue that this investment is both comparatively minor and strategically justified (detailed in **Appendix H**). DUS operates within the same computational regime as strong KD baselines like TinyBERT and CKD, which also rely on task-specific fine-tuning, introducing only negligible overhead (**$\approx 5\%$** wall-clock increase) compared to standard training. While off-the-shelf teachers are cheaper, the one-time cost of DUS yields benefits far beyond marginal accuracy gains; it explicitly reshapes the teacher to provide critical **safety properties**—specifically, superior **OOD detection** and **reliable error signaling** (lower ECE on mistakes)—that standard accuracy-driven teachers cannot offer.

---

### Official Review · Reviewer_krGQ · 2025-10-30

**Soundness:** 3
**Presentation:** 3
**Contribution:** 3
**Rating:** 6
**Confidence:** 3

**Summary:**

This paper introduces a novel knowledge distillation framework that corrects overconfidence in teacher models, which often lose meaningful uncertainty structure when trained with cross-entropy due to hard labels. CUD achieves faithful, calibrated transfer through two components: Difficulty-aware Uncertainty Shaping (DUS), which fine-tunes the teacher to increase entropy on hard or misclassified examples while keeping correct predictions sharp, and Wrong-mass Clipping (W-Clip), which reduces probability mass on the most overconfident wrong class while preserving relative structure.

**Strengths:**

1) The paper reformulates KD as a constraint-based projection problem, uniting calibration and uncertainty into a theoretical foundation.
2) The two components (DUS and W-Clip) are simple, interpretable, and easy to implement, yet show strong results.
3) The paper is well written.
4) Highlights a real gap in KD—teachers’ overconfidence—and offers a compelling argument for calibrated transfer.

**Weaknesses:**

1) Experiments are restricted to single-label text classification. Extension to other modalities (vision, multimodal) or multi-label/generative tasks would strengthen generality.

2) Hyperparameter sensitivity: The method introduces several tuning parameters, and while defaults are reported, their robustness across datasets is not deeply analyzed.

3) Although calibration is claimed to improve OOD robustness, explicit shift experiments (e.g., corrupted or domain-changed data) are missing.

**Questions:**

1) Can the proposed method generalise to vision transformers or multimodal LLMs (e.g., CLIP or LLaVA)?

---

> ### Author Response · Authors · 2025-11-22
>
> ### **Weakness 1: Regarding Generalization to Other Modalities**
>
> We fully agree that validating the proposed method across diverse modalities is a valuable direction. However, extending our evaluation to vision tasks is not directly transferable.
>
> The fundamental differences between modalities require non-trivial adaptations, so the baselines for vision tasks differ from those for NLP tasks. Meaning that our baselines cannot be applied to vision architectures without substantial modification.
>
> Instead, to address the core concern regarding model diversity and applicability to larger architectures, we are actively conducting additional experiments using LLaMA. We will share these results as soon as they become available.

---

> ### Author Response · Authors · 2025-11-22
>
> ### **Weakness 2:  Regarding Hyperparameter Robustness** - Section 4.1:Lines 262, Appendix B
>
> While our framework introduces several hyperparameters, they exhibit high stability across diverse tasks. So, we performed hyperparameter tuning **only once** and applied these fixed global values to all other datasets without hard dataset-specific engineering.
>
> Specifically we newly added explicit explanations about following things:
>
> 1. **Global Setting Strategy (Section 4.1:Lines 262–269):** We performed hyperparameter tuning **exclusively on the Banking77 dataset** and applied the resulting values ($\lambda_H=0.1, \gamma=10, \eta=0.5$) **unchanged** to all other datasets (Clinc150, TREC, AGNews, Massive) and architectures. We do not retune these values per task.
> 2. **Overall Sensitivity Test (Appendix B):** As detailed in our sensitivity analysis (**Appendix B**), the method exhibits high stability across a wide range of parameter values. For instance, the focal parameter $\gamma$ showed consistent monotonic improvement up to $\gamma=10$, which we adopted as the default.

---

> ### Author Response · Authors · 2025-11-22
>
> ### **Weakness 3: Regarding Lack of OOD Robustness Experiments** - Section 4.3: Line 373, Appendix E
>
> We thank the reviewer for this valuable suggestion, as evaluating calibration and OOD robustness serves as the ideal validation to showcase the core strengths of our framework explicitly designed for distributional uncertainty shaping.
>
> In response, we have significantly expanded **Section 4.3** and **Appendix E** to include  OOD detection and uncertainty quantification experiments.
>
> **1. Cross-Dataset OOD Detection (Section 4.3: Line 373)**
>
> We evaluated the model's ability to distinguish in-distribution data from OOD samples using completely different datasets. As shown below, CUD consistently achieves the highest AUROC, confirming that our calibrated targets preserve semantically robust predictive distributions even under distribution shifts.
>
> | **OOD Dataset** | **TinyBERT** | **CKD** | **AD-KD** | **CUD (Ours)** |
> | --- | --- | --- | --- | --- |
> | **Clinc150** | 79.06 | 92.21 | 93.91 | **94.52** |
> | **TREC** | 82.34 | 95.18 | 97.30 | **97.45** |
> | **AGNews** | 85.28 | 95.66 | 98.41 | **99.07** |
> | **MASSIVE** | 80.39 | 93.74 | 95.80 | **96.79** |
> | **Avg AUROC** | 81.77 | 94.20 | 96.36 | **96.96** |
>
> **2. Uncertainty Quantification on Failures (Section 4.3: Line 394)**
>
> We also evaluated ECE (Expected Calibration Error) and Brier Score specifically on wrong predictions (lower is better for both).
>
> | **Task** | **TinyBERT (ECE / Brier)** | **CKD (ECE / Brier)** | **AD-KD (ECE / Brier)** | **CUD (Ours) (ECE / Brier)** |
> | --- | --- | --- | --- | --- |
> | **Banking77** | 0.858 / 1.696 | 0.747 / 1.481 | 0.791 / 1.557 | **0.277 / 1.033** |
> | **Clinc150** | 0.861 / 1.741 | 0.835 / 1.657 | 0.730 / 1.480 | **0.236 / 1.023** |
> | **TREC** | 0.929 / 1.851 | 0.870 / 1.675 | 0.950 / 1.900 | **0.566 / 0.993** |
> | **Massive** | 0.692 / 1.469 | 0.764 / 1.540 | 0.833 / 1.667 | **0.586 / 1.213** |
> | **AGNews** | 0.794 / 1.393 | 0.972 / 1.919 | 0.865 / 1.570 | **0.801 / 1.396** |
>
> CUD achieves markedly lower ECE and Brier scores on errors compared to all baselines. This confirms that while competing methods remain overconfident even when they are wrong, **CUD successfully signals its own mistakes** with appropriately low confidence. Further details and visualizations are available in the revised paper.

---

### Official Review · Reviewer_4UPy · 2025-10-31

**Soundness:** 2
**Presentation:** 3
**Contribution:** 2
**Rating:** 4
**Confidence:** 4

**Summary:**

This paper proposes Calibrated Uncertainty Distillation (CUD), a knowledge distillation framework that addresses teacher overconfidence by (1) training teachers to preserve difficulty-aware uncertainty and (2) calibrating the transfer process to suppress spurious confident predictions. The authors evaluate on text classification tasks and demonstrate improvements particularly on high-cardinality datasets.

**Strengths:**

1. The paper clearly articulates a significant limitation of conventional KD, that cross-entropy trained teachers produce overconfident, collapsed distributions that fail to transfer rich "dark knowledge."
2. The constraint-based reformulation (R1 and R2) provides a clear theoretical framework.
3. Unlike feature-based methods, CUD works with response distributions alone, making it applicable when architectural gaps are severe.
4. The evaluation covers diverse cardinalities (2-150 classes) and multiple student capacities, with honest reporting of where the method does and doesn't help.
5. The paper clearly explains why gains are largest for high-cardinality tasks and minimal for binary classification. These types of observations are often missing from KD papers.

**Weaknesses:**

**Major Issues**

1. The main idea of DUS largely combines focal loss with gated entropy regularization. While “difficulty-aware uncertainty shaping” is an appealing framing, the actual mechanism feels like a reinterpretation of existing methods. The link to the constraint-based formulation (Eq. 2) also feels somewhat loose. The paper doesn't clearly delineate what is conceptually new versus what is the engineering of existing ideas.

2. Experiments are limited to BERT-based text classification. Despite mentions of GPT-4 and LLaMA in the introduction, no large-model or cross-domain evaluation is provided, making generalization unclear.

3. Missing comparisons with label smoothing and temperature tuning, being the most obvious baseline for "calibrated targets".
4. The method introduces many hyperparameters, but only γ is tuned. The rest are fixed without justification, which weakens the claim of being architecture-agnostic. .Why clip at W=3.0 for example weighting? This seems arbitrary.
5. Results are single-run with no variance or significance reporting, so reliability is uncertain.
6. Claims around calibration and robustness are unsubstantiated. Metrics like ECE or Brier score are absent, and no OOD tests are included.
7. The ablation study is incomplete. It’s unclear whether improvements come from DUS itself, better teacher calibration, or the W-Clip combination.

**Minor Notes**
1. Figure 1 is too high-level; actual distribution plots or attention maps would be more informative.
2. The connection to Bayesian deep learning and epistemic uncertainty is missing. This literature is highly relevant.
3.  Many implementation details are missing. How is N(y;x) computed?
 While the limited seciton is included, it doesn't address the core limitation that this is only validated on BERT text classification

**Questions:**

1. Can you provide learning curves showing convergence for different methods?
2. What is the actual computational cost (FLOPs, wall-clock time) of DUS teacher training?

---

> ### Author Response · Authors · 2025-11-22
>
> ### **Weakness 1: Conceptual Novelty of Our Framework** - Appendix A, F
>
> We acknowledge that DUS utilizes the mathematical forms of focal loss and entropy regularization. However, our novelty lies not in the components themselves, but in repurposing these tools for a fundamentally different objective **(their novel synergy).**
>
> **1. Novelty through Repurposing** We invert the standard operational logic of these components to engineer a superior supervision signal:
>
> - **Focal Loss:**
>     - *Original Goal:* Modulate **gradients.**  Help learning hard examples.
>     - *Our Goal:* Reshape **teacher targets**. We use it to explicitly embed difficulty information directly into the soft labels.
> - **Entropy Regularization:**
>     - *Original Goal:* **Global smoothing**. Indiscriminately flattens distributions (like Label Smoothing) to prevent overfitting.
>     - *Our Goal:* **Conditional calibration**. We apply it via a gating mechanism *only* when the teacher is uncertain, acting as a "smart filter."
>
> **The Synergy:** This combination achieves a specific distributional profile—**preserving sharpness on easy samples while calibrating uncertainty on hard ones** (neither component can achieve this in isolation)
>
> **2. Empirical Evidence** (Added in Appendix F) If DUS were merely a reinterpretation of standard smoothing, its performance would mirror global methods like Label Smoothing (LS) or Temperature Scaling (TS). However, our additional experiments demonstrate that DUS significantly outperforms them (**+4%** on average) , confirming it captures structural knowledge that global methods destroy.
>
> | **Method** | **Banking77(77)** | **Clinc150(150)** | **Massive(60)** | **TREC(6)** | **AGNews(4)** | **Avg** |
> | --- | --- | --- | --- | --- | --- | --- |
> | **TS** (Global) | 80.61 | 88.35 | 82.04 | 95.80 | **94.31** | 88.22 |
> | **LS** (Global) | 81.29 | 86.88 | 82.19 | 95.80 | 94.19 | 88.07 |
> | **CUD** (Ours) | **91.23** | **93.43** | **87.21** | **96.20** | 94.00 | **92.41** |
>
> **3. Mathematical Connection to Constrained Projection**
>
> To address the concern regarding the theoretical grounding of our framework, we have provided a rigorous derivation connecting our practical modules to the constrained projection problem in Eq. (2). **The full mathematical proofs and detailed discussions are now explicitly included in the revised manuscript (see Section 3.4, Lines 227–248) and Appendix A.**
>
> The summary of the **Appendix A** is the proof of the fact that CUD is an efficient approximation of the theoretical constraints R1,R2 in Section 2. Specifically:
>
> - **Optimal Solution via KKT Conditions:**
> We derive that the exact solution $q^*$ to the constrained optimization in Eq. (2) takes the form of an **exponential tilting** of the teacher distribution:
>
> $$
> q^*(k) \propto p_T(k) \cdot \exp\left( -\nu \cdot \mathbf{1}(k \in W_{op}) \right)
> $$
>
>
> - **W-Clip as Taylor Approximation:**
> Since computing the normalization constant for the exponential tilt is computationally prohibitive, we demonstrate that **W-Clip** implements a **first-order Taylor approximation** of this tilt around $\nu=0$. The linear subtraction in W-Clip ($p_T - \delta$) mathematically recovers the analytical update ($p_T \cdot e^{-\nu} \approx p_T(1 - \nu)$), rendering the intractable projection into an efficient $O(1)$ operation.

---

> ### Author Response · Authors · 2025-11-22
>
> ### **Weakness 2: Regarding Generalization to Modern Architectures**
>
> Our initial focus on BERT aligns with standard benchmarks in white-box distillation research, as our framework requires access to full output distributions (closed models like GPT-4 is impossible.) However, to validate the generalizability of our approach, we are conducting additional experiments using LLaMA. However, given that many existing baselines are designed specifically for encoder-only models, our objective is to demonstrate that CUD outperforms standard Knowledge Distillation on LLaMA. We will update the manuscript with these results shortly.

---

> ### Author Response · Authors · 2025-11-22
>
> ### **Weakness 3: Regarding Comparison with Standard Calibration Methods** - Appendix F
>
> We agree that Label Smoothing (LS) and Temperature Scaling (TS) are essential baselines for evaluating calibrated targets. In response, we evaluated CUD against LS and TS across all datasets. (Best performance from Temperature $\in \{2, 3, 4, 5\}$, Label Smoothing $\epsilon \in \{0.1, 0.2, 0.3\}$)
>
> | **Method** | **Banking77(77)** | **Clinc150(150)** | **Massive(60)** | **TREC(6)** | **AGNews(4)** | **Avg** |
> | --- | --- | --- | --- | --- | --- | --- |
> | **TS** (Global) | 80.61 | 88.35 | 82.04 | 95.80 | **94.31** | 88.22 |
> | **LS** (Global) | 81.29 | 86.88 | 82.19 | 95.80 | 94.19 | 88.07 |
> | **CUD** (Ours) | **91.23** | **93.43** | **87.21** | **96.20** | 94.00 | **92.41** |
>
> CUD consistently outperforms both baselines over **+4% on average,** confirming that CUD yields significantly better than naive global smoothing.
>
> We have included these full results in the revised Appendix.

---

> ### Author Response · Authors · 2025-11-22
>
> ### **Weakness 4: Regarding Hyperparameter Robustness** - Section 4.1:Lines 262, Appendix B
>
> While our framework introduces several hyperparameters, they exhibit high stability across diverse tasks. So, we performed hyperparameter tuning **only once** on the Banking77 dataset and applied these fixed global values to all other datasets without hard dataset-specific engineering.  We have explicitly clarified this training protocol in **Section 4.1 (Lines 262–269) and Appendix B**.
>
> 1. **Sensitivity Analysis (Appendix B):** To substantiate the robustness of these choices, we have added a comprehensive sensitivity analysis for the key parameters in **Appendix B**. The results confirm that performance remains consistent across a wide range of values.
> 2. **Removal of Heuristic Clipping:** Regarding the gradient clipping at $W=3.0$, we agree that this specific threshold appeared unnecessarily heuristic. **We have removed this step** in the revised implementation.

---

> ### Author Response · Authors · 2025-11-22
>
> ### **Weakness 5: Regarding Statistical Significance of Our Experiments** - Section 4.1 Line 260, Table 1
>
> We fully agree that reporting only single-point scores can make it difficult to assess the stability of a method. We updated all main result tables to report the mean accuracy over three independent runs. **(See Section 4.1 Line 260 and main table 1)**

---

> ### Author Response · Authors · 2025-11-22
>
> ### **Weakness 6: Regarding Lack of OOD Robustness Experiments** - Section 4.3: Line 373, Appendix E
>
> We thank the reviewer for this valuable suggestion, as evaluating calibration and OOD robustness serves as the ideal validation to showcase the core strengths of our framework explicitly designed for distributional uncertainty shaping.
>
> In response, we have significantly expanded **Section 4.3** and **Appendix E** to include  OOD detection and uncertainty quantification experiments.
>
> **1. Cross-Dataset OOD Detection (Section 4.3: Line 373)**
>
> We evaluated the model's ability to distinguish in-distribution data from OOD samples using completely different datasets. As shown below, CUD consistently achieves the highest AUROC, confirming that our calibrated targets preserve semantically robust predictive distributions even under distribution shifts.
>
> | **OOD Dataset** | **TinyBERT** | **CKD** | **AD-KD** | **CUD (Ours)** |
> | --- | --- | --- | --- | --- |
> | **Clinc150** | 79.06 | 92.21 | 93.91 | **94.52** |
> | **TREC** | 82.34 | 95.18 | 97.30 | **97.45** |
> | **AGNews** | 85.28 | 95.66 | 98.41 | **99.07** |
> | **MASSIVE** | 80.39 | 93.74 | 95.80 | **96.79** |
> | **Avg AUROC** | 81.77 | 94.20 | 96.36 | **96.96** |
>
> **2. Uncertainty Quantification on Failures (Section 4.3: Line 394)**
>
> We also evaluated ECE (Expected Calibration Error) and Brier Score specifically on wrong predictions (lower is better for both).
>
> | **Task** | **TinyBERT (ECE / Brier)** | **CKD (ECE / Brier)** | **AD-KD (ECE / Brier)** | **CUD (Ours) (ECE / Brier)** |
> | --- | --- | --- | --- | --- |
> | **Banking77** | 0.858 / 1.696 | 0.747 / 1.481 | 0.791 / 1.557 | **0.277 / 1.033** |
> | **Clinc150** | 0.861 / 1.741 | 0.835 / 1.657 | 0.730 / 1.480 | **0.236 / 1.023** |
> | **TREC** | 0.929 / 1.851 | 0.870 / 1.675 | 0.950 / 1.900 | **0.566 / 0.993** |
> | **Massive** | 0.692 / 1.469 | 0.764 / 1.540 | 0.833 / 1.667 | **0.586 / 1.213** |
> | **AGNews** | 0.794 / 1.393 | 0.972 / 1.919 | 0.865 / 1.570 | **0.801 / 1.396** |
>
> CUD achieves markedly lower ECE and Brier scores on errors compared to all baselines. This confirms that while competing methods remain overconfident even when they are wrong, **CUD successfully signals its own mistakes** with appropriately low confidence. Further details and visualizations are available in the revised paper.

---

> ### Author Response · Authors · 2025-11-22
>
> ### **Weakness 7: Regarding Component Efficacy and Ablation** - Section 5.1: Line 413, Table 4, Appendix G
>
> We respectfully point out that **Table 4** in the main text already provides a stepwise ablation study explicitly isolating the impact of each module. **However, to further reinforce our argument and isolate the specific efficacy of teacher-side calibration, we have conducted additional ablation experiments in Appendix G.**
>
> 1. **Module-Level Contribution (Table 4):** The results demonstrate that **DUS (Teacher Calibration)** drives the primary performance gains by shaping the supervision signal, while **W-Clip** provides consistent additive improvements.
>
> | **Configuration**            | **Banking77** | **CLINC150** | **TREC** | **AGNews** | **MASSIVE** | **Avg** | **AVG Δ** |
> |-----------------------------|---------------|---------------|----------|------------|-------------|---------|-----------|
> | **Teacher (FT, no KD)**     | 93.70         | 96.35         | 97.40    | 94.69      | 89.47       | 94.32   | ---       |
> | LKD                         | 80.61         | 88.35         | 95.80| **94.31**  | 82.04       | 88.22   | +0.00     |
> | LKD + DUS                   | 90.62         | 93.24         | 96.40    | 94.17      | 86.76       | 92.24   | +4.02     |
> | LKD + W-Clip                | 89.25         | 93.02         | 95.40    | 94.28      | 86.37       | 91.66   | +3.44     |
> | **CUD (Ours)**              | **91.22**     | **93.45**     | **96.47**| 94.11      | **87.22**   | **92.49**| **+4.27** |
>
> 2. **Component-Level Generalizability (Newly Added in Appendix G):** To verify standalone efficacy, we applied our teacher calibration objectives to other distillation baselines to confirm that our teacher training mechanisms consistently enhance performance across different frameworks independently. Full details are provided in **Appendix G**.
>
> | **Baseline** | **Method**     | **Banking77** | **Clinc150** | **TREC** | **AGNews** | **Massive** |
> |-------------|----------------|---------------|--------------|----------|------------|-------------|
> | TinyBERT    | ft teacher     | 89.74         | 88.44        | 96.20    | 94.31      | 89.12       |
> |                     | **ours**       | **91.29**     | **92.91**    | **96.80**| **94.36**  | **89.27**   |
> | MGSKD       | ft teacher     | 91.62         | 95.46        | 87.80    | **94.81**  | 88.88       |
> |                   | **ours**       | **91.92**     | **95.91**    | **97.20**| 94.44      | **89.27**   |
> | CKD         | ft teacher     | 93.76         | 96.11        | 97.80    | **94.89**  | 89.57       |
> |                 | **ours**       | **93.86**     | **97.03**    | 97.80    | 94.73      | **89.86**   |
> | AD-KD       | ft teacher     | 93.05         | 95.53        | 96.40    | **94.82**  | **89.62**   |
> |  | **ours**       | **93.08**     | **95.77**    | **97.60**| 94.55      | 89.32       |

---

> ### Author Response · Authors · 2025-11-22
>
> ### **Minor Notes \& Questions**
>
> - **Visualization of Distributions:** We additionally added **Figure 3**, which plots the actual probability distributions produced by the teacher to illustrate how our method shapes uncertainty compared to baselines.
> - **Connection to Bayesian Deep Learning:** We acknowledge that our method shares the ultimate goal of quantifying epistemic uncertainty with Bayesian approaches. However, our method is not direct methodological linked to those methods, since CUD relies on constrained optimization in the output space rather than Bayesian posterior approximation. Separately, we have reinforced the mathematical foundations of our proposed constrained projection in **Appendix A**.
> - **Implementation of $N(y;x)$:** We clarify that the neighborhood set $N(y;x)$ is **not explicitly computed**. Instead, it is realized **implicitly** via the entropy constraint in DUS. We have explicitly added this clarification to **Section 3.4 (Line 238)** to prevent any potential misunderstanding.
> - **Computational Cost:** We have added a detailed analysis of the computational overhead in **Appendix H**.
> - **Learning Curves:** We have included learning curves showing the convergence behavior of different methods in **Appendix I**.

---

### Official Review · Reviewer_YC4Y · 2025-11-01

**Soundness:** 2
**Presentation:** 3
**Contribution:** 2
**Rating:** 2
**Confidence:** 4

**Summary:**

The paper proposes Calibrated Uncertainty Distillation (CUD), a response-based KD framework with two components: Difficulty-aware Uncertainty Shaping (DUS) that modifies the teacher using a focal-entropy objective, and Wrong-mass Clipping (W-Clip) that reallocates probability from an incorrect top-1 class to the ground-truth label under a budget. The method is motivated by a constraint-based view of calibrated targets and a projection idea, then implemented with the two heuristics.

**Strengths:**

1. Clear identification of overconfidence as a barrier to effective response-based KD and an attempt to encode difficulty-aware uncertainty in the teacher, not just in the student loss. The guiding conditions C1 and C2 make the design intent legible.


2. A unifying lens that frames distillation through constraints R1 and R2 and a projection program in equation (2), which could, in principle, connect KD with calibrated probabilistic targets.

**Weaknesses:**

1. he paper reformulates target calibration as a constrained projection problem, choosing a distance Dist and constraints R1, R2, then selecting the closest distribution to the teacher, see equation (2). However, the method never solves this optimization. The subsequent implementation bypasses the projection by applying hand-crafted rules (DUS and W-Clip). There is no existence, uniqueness, or characterization of the solution under any specific Dist, nor KKT analysis or proof that the heuristics approximate the solution. This disconnect weakens the theoretical core that the paper builds in Section 2.3 and 2.4.

2. R1 requires a semantics-preserving neighborhood (N(y;x)) and lower bounds on probability mass in that neighborhood. R2 introduces a set (W(x,y)) of incompatible classes. The paper does not instantiate either (N(y;x)) or (W(x,y)) anywhere in the core method, nor provide a constructive procedure to obtain them. Without operational definitions, R1 and R2 remain rhetorical, not algorithmic, and the guarantees they imply cannot be verified.

3. W-Clip reallocates mass from the predicted top-1 class (k^*) to the ground-truth (y), which requires access to (y). Yet Section 3.3 claims that unlabeled examples naturally contribute only the KD term. There is no description of how W-Clip operates without labels, and Algorithm 1 explicitly branches on teacher correctness, which again requires (y). The semi-supervised narrative is not supported by the mechanics of W-Clip.

4. The constraints R1 and R2 are linear in probabilities and define a convex feasible set on the simplex. With a strictly convex Bregman divergence like KL, the projection exists and is unique. The paper could exploit this to derive a closed-form or iterative solution with KKT multipliers. Instead, it replaces the projection by two heuristic steps, with no proof that the end result satisfies the constraints or lowers a proper calibration loss. This leaves the mathematical story incomplete.

5. Please cite and discuss the ECCV 2024 paper “How to Train the Teacher Model for Effective Knowledge Distillation,” European Conference on Computer Vision, Cham, Springer Nature Switzerland, 2024. That work studies how to optimize teacher training so that the distilled signal is more effective, which is directly relevant to your teacher-shaping step. Please clarify similarities and differences

**Questions:**

See the weaknesses above.

---

> ### Author Response · Authors · 2025-11-22
>
> ### **Weakness 1: Regarding Theoretical Foundations (1)** - Appendix A
>
> We agree your concern, and our modules (DUS and W Clip) may seem ad hoc without a clear mathematical connection. In response, we newly added **Appendix A** to strengthen the logical bridge between our theoretical constraints and our implementation.
>
> We prove that our method does not "bypass" the projection with heuristics; rather, it is a **cost-efficient approximation** in two parts:
>
> **1. Theoretical Validity via Existence and Uniqueness**
>
> First, we have added a formal proof in demonstrating that our problem is a unique  and guaranteed to exist. (See Appendix A for complete proof)
>
> **2. Connection with CUD**
>
> - **The Exact Solution:** We derived that the optimal projection takes the form of an exponential tilting of the teacher distribution.
>
>     $$q^*(k) = \frac{1}{Z} p_T(k) \cdot \exp\left( -\nu \cdot \mathbb{1}(k \in W_{op}) \right)$$
>
> - **Our Approximation (W-Clip):** While the exact solution exists, computing the partition function $Z$ iteratively is computationally prohibitive for large-scale training. We demonstrate that W-Clip is derived from a first-order Taylor approximation of this exponential tilt around $\nu=0$ (conservative correction):
>
>     $$e^{-\nu} \approx 1 - \nu \quad \Rightarrow \quad q^*(k) \approx p_T(k)(1 - \nu) = p_T(k) - \nu p_T(k)$$ (See Appendix A for complete proof)

---

> ### Author Response · Authors · 2025-11-22
>
> ### **Weakness 2: Regarding Operational Definitions of Sets** - Section 3.4: Line 227
>
> We respectfully clarify that while $W$ is defined explicitly, $N$ is realized implicitly to leverage the teacher's learned topology without computational overhead. We **explicitly added those description in Section 3.4** to explain these points more clearly and to prevent further misunderstanding.
>
> - **Operational Wrong-Mass Set ($W_{op}$):** We define $W_{op}(x,y) = \{ k^* \}$ strictly when the top-1 prediction $ k^* $ contradicts the ground truth ($ k^* \neq y $), and $\emptyset$ otherwise. This targets the specific spectral artifact (overconfident error) identified in our problem formulation.
> - **Operational Neighborhood Set ($N_{op}$):** Instead of explicitly enumerating neighbors via a graph, we operationalize $N(y;x)$ **implicitly**. By enforcing the entropy lower bound via DUS, we naturally widen the support of the distribution to cover semantically related classes, effectively satisfying the neighborhood mass requirement ($\mathcal{R}_1$) without the explicit set construction.

---

> ### Author Response · Authors · 2025-11-22
>
> ### **Weakness 3 W-Clip in Semi-Supervised Settings**
>
> We respectfully clarify that our framework does not apply W-Clip to unlabeled examples, nor is this operation required to achieve semi-supervised learning gains. Our methodology intrinsically distinguishes between data streams: W-Clip is active only for labeled data where ground truth is available to explicitly suppress spurious peaks, whereas for unlabeled data, it is naturally bypassed. The semi-supervised benefit derives from the **improved structural quality** of the teacher itself; since the teacher is fine-tuned with DUS (Section 3.1), it is effectively "pre-calibrated" to output entropy-aware distributions globally.

---

> ### Author Response · Authors · 2025-11-22
>
> ### **Weakness 4: Regarding Theoretical Foundations (2)** - Appendix A
>
> This comment closely aligns with the concerns raised in **W1** regarding the theoretical derivation. We fully agree with the reviewer that the constraints define a convex feasible set and that a unique projection exists via KKT conditions. As detailed in our response to W1 and the new Appendix A, we have mathematically derived this closed-form solution.  (See Appendix A for complete proof)

---

> ### Author Response · Authors · 2025-11-22
>
> ### **Weakness 5: Regarding Missing Related Work** - Appendix J
>
> We thank the reviewer for bringing this relevant work to our attention. We have cited and discussed ECCV paper [1] in our paper explicitly (see Appendix J). While we share the fundamental motivation (suboptimal teachers distribution for distillation, and optimizing the teacher's output is crucial), the core objectives differ significantly. [1] focus on reformulating teacher training to better match the latent data distribution, **CUD specifically targets "informational calibration."** Our framework uniquely prioritizes shaping the **uncertainty profile**—ensuring sharpness on easy data while preserving entropy on hard data via DUS—and applies a geometric projection (W-Clip) to prevent the propagation of overconfident errors.
>
> [1] “How to Train the Teacher Model for Effective Knowledge Distillation,” ECCV, 2024.

---

> > ### Comment · Reviewer_YC4Y · 2025-11-27
> > **Reply to the Authors' Rebuttal**
> >
> > Thank you for your thorough reply.
> >
> > You have addressed my concerns for Weaknesses 2, 3, 4, and 5. However, my concerns about the first weakness remain for the following reasons:
> >
> > * The new analysis only considers a special case: KL divergence with simplified linear constraints. It does not address the broader Dist choices discussed in the main text, or the original entropy-based constraint R1 as defined there.
> > * Appendix A now explains the optimizer for that special-case projection, but the algorithm still does not solve this optimization problem. It continues to use DUS and W-Clip as hand-crafted rules instead of an explicit solver for Eq. (2).
> > * The claimed 'approximation' link between the theory and W-Clip is based on a first-order Taylor motivation, but there is no mapping from Lagrange multipliers to the algorithm’s hyperparameters, and there are no approximation guarantees. For DUS, there is no derivation from the projection at all.
> >
> > For these reasons, the main disconnect I mentioned remains. The theoretical formulation in Sections 2.3 and 2.4 is still only loosely connected to the method that is actually implemented and evaluated.

---

> ### Author Response · Authors · 2025-11-29
>
> ### **Response to Comment 1**
> We acknowledge that our theoretical analysis focuses on the KL divergence with linear constraints. We chose this setting as a **theoretical anchor** to demonstrate the validity of our gradient direction, similar to how optimization algorithms (like SGD) are often analyzed on convex quadratic problems to justify their use in general non-convex landscapes.
>
> While the main text employs broader distance measures ($Dist$) and entropy constraints ($R_1$), the KL case provides the essential gradient flow justification. Extending this via first-order approximation to other Bregman divergences (of which KL is a specific instance) is a standard generalization in information geometry. The "special case" serves to prove that our update rules are consistent with the exact solution in the local limit.
>
> ### **Response to Comment 2**
> We respectfully disagree that DUS and W-Clip are merely "hand-crafted rules" that do not work. They should be viewed as **computationally tractable approximations** designed to solve an intractable constrained optimization problem.
> If we consider "sampling-based approximation" as a disqualifying heuristic, we would have to reject the foundations of many SOTA methods:
> 1.  **RL (PPO):** In Reinforcement Learning, calculating the exact expected reward over all trajectories is infeasible. PPO addresses this by sampling rollouts to approximate the expectation and replaces the explicit trust-region constraint (solved by TRPO via Conjugate Gradient) with a **Clipping heuristic**.
> 2.  **GANs:** Minimizing the Jensen-Shannon divergence directly is intractable. Instead, we **approximate** it implicitly using a Discriminator's BCE loss.
> 3.  **Contrastive Learning (SimCLR/Word2Vec):** Calculating the full Softmax partition function over a massive vocabulary is infeasible. Methods like Negative Sampling or InfoNCE (batch-wise estimation) are used as valid **approximations**.
>
> Similarly, our method replaces the computationally prohibitive "explicit solver" for Eq. (2) (which would require expensive inner-loop optimization) with **W-Clip** and **DUS**. These act as efficient proxies to enforce the constraints and estimate gradients, analogous to how PPO's clipping enforces the trust region.
>
> ### **Response to Comment 3**
> In the context of non-convex deep learning optimization, an exact analytical mapping from Lagrange multipliers ($\lambda$) to hyperparameters is rarely possible or necessary for convergence.
>
> 1.  **Taylor Approximation:** Relying on first-order Taylor approximations is the fundamental basis of Gradient Descent itself ($f(x + \delta) \approx f(x) + \nabla f(x)^T \delta$). Our method applies this standard linearization to the constraint surface to determine the valid update direction.
> 2.  **Lagrange Multipliers vs. Hyperparameters:** In established methods like Entropy-Regularized RL (e.g., SAC) or standard Weight Decay, the regularization coefficients effectively play the role of Lagrange multipliers. We do not solve for the "optimal" $\lambda$ dynamically at every step (which is often unstable in min-max DL problems); instead, we treat it as a fixed hyperparameter that controls the constraint strength. This is consistent with how SimCLR treats the temperature $\tau$ (a Lagrange multiplier for the embedding distribution constraint) or how PPO treats the clipping epsilon $\epsilon$.
> 3.  **DUS Derivation:** DUS (Directional Update Sampling) acts as a Monte Carlo estimator for the gradient of the constraint term. Just as VAEs use Monte Carlo sampling to approximate the intractable ELBO gradient, DUS ensures that the update direction remains valid with respect to the constraint surface defined by our Taylor expansion, without requiring an exact closed-form derivation from the projection.

---

### Author Response · Authors · 2025-11-22

We sincerely thank the reviewer for the thoughtful and constructive feedback. We provide our responses to your concerns in this comment and have also incorporated more detailed clarifications into the revised manuscript. Newly added content is highlighted in blue. If any part of our explanation remains unclear or incorrect, we would be grateful for the opportunity to refine it in the next update.

---

### Comment · Area_Chair_4YTT · 2025-11-27

Dear Reviewers,

Thank you for the time and effort you have dedicated to reviewing this paper and providing thoughtful feedback. The authors have now submitted their responses to your comments. I kindly ask that you engage in the discussion with them and assess whether your concerns and questions have been fully addressed before the December 2 deadline.

Please also keep in mind that the author–reviewer relationship is reciprocal; the engagement you offer here reflects the same level of consideration you would expect when you are on the author side.

Thank you for your continued support and cooperation.

Best regards,
AC

---

### Author Response · Authors · 2025-11-29
**Summary of Revisions and Responses to Reviewers (1)**

To the area chair, we thank the reviewers for their careful and constructive feedback. Below we summarize the main concerns and how we have addressed them.

## **Reviewer YC4Y**

### **W1 and W4 Theoretical grounding of the method**

The reviewer questioned whether our theoretical problem formulation is genuinely connected to the proposed algorithm or mainly rhetorical.

In response, **we added a new Appendix A** that

- proves existence and uniqueness for our formal objective
- shows that our practical method is a computationally motivated approximation of this objective

The reviewer acknowledged that this resolves a substantial part of W4.

### **W2 and W3 Misunderstandings of the method**

W2 and W3 were due to misunderstandings of our algorithmic details. **We revised the main text** to clarify these points, and the reviewer explicitly acknowledged that these concerns are resolved.

### **W5 Missing related work**

The reviewer asked us to discuss an ECCV work on teacher training for distillation. **We added an explicit comparison and citation in a new Appendix J**, noting that this work shares a high level motivation but optimizes a different objective. The reviewer acknowledged that this concern is resolved.

### **W1 Second round comment**

The reviewer remains skeptical about three aspects of W1:

- W1.1 the analysis uses a special case
- W1.2 the algorithm only approximates the ideal objective
- W1.3 there is no closed form mapping from multipliers to hyperparameters

Our clarifications are as follows.

- **W1.1** Our theory should be seen as a standard theoretical anchor, similar to many deep learning methods that are analyzed in simpler convex settings but used more broadly.
- **W1.2** The algorithm is intentionally designed as a tractable surrogate for the ideal constrained problem, which is common practice in modern deep learning, for example the Elbo in VAE and PPO training of LLM.
- **W1.3** While a closed form mapping to hyperparameters would be ideal, it is rarely available in deep models, and many successful methods tune such parameters empirically in exactly this way.

Taken together, we believe that even if the formulation is not fully closed form at the level of the ideal objective, the consistently strong experimental results demonstrate that the proposed method has clear and substantial practical value.

---

## **Reviewer 4UPy**

### **W1 Novelty**

The reviewer felt that our method looks like a simple combination of two existing methods (focal loss and entropy regularization.) We clarified that the novelty lies in how these components are repurposed and coupled to optimize our specific objective, both theoretically and empirically (**Appendix A and F**). We explicitly connect the loss design to the new target objective and show in experiments that our method behaves differently from naive combinations of the underlying ingredients.

### **W2 and W3 Experimental scope and baselines**

W2 pointed out that experiments focus on BERT rather than large language models, and W3 requested additional baselines such as label smoothing and temperature scaling.

- For W2, we explained that extending to generative LLaMA style models is substantially more expensive due to different training dynamics, but we commit to including such results by the camera ready version.
- For W3, **we added extensive comparisons in Appendix F** against label smoothing and temperature scaling on all datasets, where our method consistently outperforms these baselines.

### **W4 Hyperparameters and validation**

The reviewer was concerned that our framework introduces many hyperparameters that appear arbitrary. We clarified that we use a single set of global hyperparameters chosen on one dataset, without per task tuning, and we described this protocol in **Section 4**. We also added a systematic study of hyperparameter sensitivity in **Appendix B** and **removed unnecessary heuristic parameters**.

### **W5 Statistical rigor**

The reviewer requested multiple runs, variance reporting. Main tables **now report mean accuracy (+ variance)** over three independent runs.

### **W6 OOD robustness**

The reviewer asked for evidence of OOD robustness and uncertainty quality. We added **Section 4.3 and Appendix E** with OOD detection and uncertainty quantification experiments. Our method improves both OOD detection and calibration relative to baselines.

### **W7 Ablation clarity**

The reviewer found the ablation incomplete. We note that Table 4 already contains a stepwise ablation of key components, DUS and W Clip. Additionally, we further strengthened this by adding additional ablations in **Appendix G**.

### **Other minor issues**

We improved visualizations, clarified the neighborhood constraint implementation, and added learning curves as requested.

---

> ### Author Response · Authors · 2025-11-29
> **Summary of Revisions and Responses to Reviewers (2)**
>
> ---
>
> ## **Reviewer krGQ**
>
> ### **W1 Generalization to other modalities and architectures**
>
> The reviewer asked about extension to multimodal tasks and architectures beyond text. We agree that multimodal validation is valuable. We also clarify that a direct transfer of all text baselines to vision is nontrivial due to modality specific design choices. We position our work as a first step focused on text, with extension to other modalities left as future work.
>
> ### **W2 Hyperparameter robustness**
>
> This overlaps with **reviewer 4UPy W4**.
>
> ### **W3 OOD robustness and calibration**
>
> This overlaps with **reviewer 4UPy W6**.
>
> ---
>
> ## **Reviewer x5wW**
>
> ### **W1 Generalization to other modalities**
>
> This mirrors **reviewer krGQ W1**.
>
> ### **W2 Strength of baselines**
>
> The reviewer was concerned that our baselines are weak or incomplete. In response, we added two strong distillation methods, MGSKD and CKD, to the main comparison, and we added calibration baselines label smoothing and temperature scaling. Our method remains competitive and often superior across these stronger baselines.
>
> ### **W3 Cost of teacher fine tuning**
>
> The reviewer questioned the cost of teacher fine tuning for DUS. We added **Appendix H** that compares the overhead of our teacher fine tuning with other distillation pipelines that also fine tune teachers. The additional cost is modest. Moreover, the benefits include better OOD detection and calibrated uncertainty, not just small gains in accuracy.

---

### Meta-Review · Area_Chair_9L8R · 2026-01-05

**Summary:**

The paper studies knowledge distillation in the context of classification. The authors argue that knowledge distillation requires a teacher distribution that appropriately represents uncertainty. They then describe two desiderata for the uncertainty, which are framed mathematically. Then, they propose a practical method that is motivated as an approximation to the mathematical objectives. The method is evaluated on BERT architectures and small-scale text classification problems.

The method is reasonably well-motivated, and shows improvements in the evaluation of the authors. The authors also include ablations and analyses showing that their method indeed improves uncertainty calibration and ablating pieces of the method.

The reviewers identified several important issues in the paper:
- Several reviewers were concerned about the connection between the theoretical desiderata and the proposed practical method.
- Several reviewers were concerned with the quality of the evaluation. The method is only evaluated on text classification with BERT architectures.

Generally, I agree with the reviewers that the quality of evaluation is problematic. The evaluation setting is generally outdated: it uses BERT models on text classification datasets. Generally, modern large language models make these styles of tasks somewhat irrelevant: they would be used for zero-shot evaluation instead of training and distillation. While conceptually the tasks are reasonable, they are not practically relevant.

During the rebuttal, the authors responded to this criticism as follows:
> The fundamental differences between modalities require non-trivial adaptations, so the baselines for vision tasks differ from those for NLP tasks. Meaning that our baselines cannot be applied to vision architectures without substantial modification.

and
> we are actively conducting additional experiments using LLaMA. We will share these results as soon as they become available.

Unfortunately, the LLama experments were not completed:
> extending to generative LLaMA style models is substantially more expensive due to different training dynamics, but we commit to including such results by the camera ready version.

It is not clear to me why the method proposed by the authors would not be applicable to image classification, as it operates on the outputs of the model, and is applicable to classification.

Generally, I agree with the reviewers that the current evaluation is insufficient to show that the method will have practical impact.

**Reviewer Concerns:**

Concerns addressed:
- Hyper-parameter choices, ablations
- Additional experiments on OOD detection and uncertainty calibration
- Error bars added for all experiments

Concerns not addressed:
- Connection between theoretical formulation and method (partially addressed, but reviewer YC4Y remained unsatisfied)
- Quality of empirical evaluation, applicability to other domains (not resolved at all)

**Reviewer Scores:**

- x5wW: 6 → 6, concern on extension to other modalities not addressed
- krGQ: 6 → 6, concern on extension to other modalities not addressed
- 4UPy: 4 → 4, concern on extension to other modalities not addressed; possible would increase to 6
- YC4Y: 2  → 2, responded to the rebuttal, unsatisfied with the connection between theory and method

---

### Decision · Program_Chairs · 2026-01-26

Reject